# Block Rotation is All You Need for MXFP4 Quantization

Yuantian Shao [1 2]  Peisong Wang[✉ 2 3]  Yuanteng Chen [2 3 4]
Chang Xu [5]  Zhihui Wei[✉ 1]  Jian Cheng[✉ 2 3 4]

## Abstract

Large language models (LLMs) have achieved remarkable success, but their rapidly growing scale imposes prohibitive costs in memory, computation, and energy. Post-training quantization (PTQ) is a promising solution for efficient deployment, yet achieving accurate W4A4 quantization remains an open challenge. While most existing methods are designed for INT4 formats, the emergence of MXFP4—a new FP4 format with various hardware support (NVIDIA, AMD, Intel)—raises questions about the applicability of current techniques. In this work, we present a unified empirical comparison of representative PTQ methods under the MXFP4 format. Through systematic evaluation, we find that methods like GPTQ consistently deliver strong performance, whereas rotation-based approaches, which are widely used in state-of-the-art approaches, suffer from severe incompatibility with MXFP4. We further provide the first in-depth analysis of this conflict, tracing its root to a fundamental mismatch between MXFP4's PoT (power-of-two) block scaling and the redistribution of outlier energy via global rotation. Building on this insight, we propose a simple yet effective block rotation strategy that adapts rotation-based methods to MXFP4, leading to substantial accuracy improvements across diverse LLMs. Our findings not only offer clear guidance for practitioners but also set a foundation for advancing PTQ research under emerging low-precision formats.

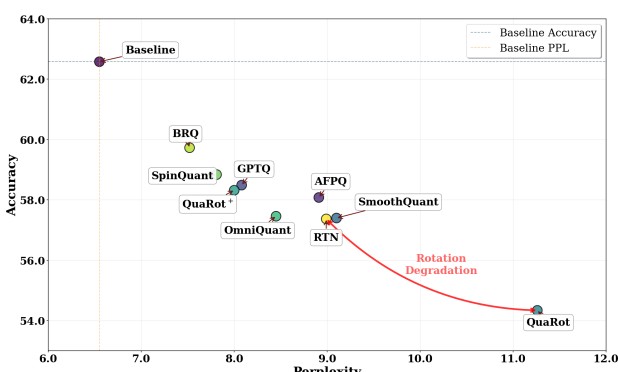

*Figure 1.* Overall performance of quantization methods under MXFP4. The x-axis shows perplexity, the y-axis shows average downstream accuracy, and methods nearer the top-left are closer to the FP16 baseline, indicating better performance.

## 1. Introduction

Large language models (LLMs) have become the cornerstone of modern artificial intelligence, but their ever-increasing scale incurs substantial memory, computation, and energy costs (Wu et al., 2025; Jin et al., 2025). Among numerous model compression techniques, post-training quantization (PTQ) has emerged as a practical solution due to its training-free nature and low engineering overhead (Czakó et al., 2025; Yang et al., 2024). While INT8 and INT4 quantization have already been adopted in practice, achieving accurate W4A4 (4-bit weights and 4-bit activations) remains a critical challenge (Elangovan et al., 2025). For recently released LLMs (e.g., LLaMA-3.2 1B/3B), naive 4-bit quantization often results in severe performance degradation (van Breugel et al., 2025), making W4A4 a key research frontier for efficient LLM deployment.

Meanwhile, hardware advances have spurred the microscaling (MX) family of data formats (Han et al., 2025), such as MXFP4. MXFP4 is an open standard format proposed by the Open Compute Project (OCP) (Rouhani et al., 2023), and is currently supported by AMD Ryzen AI MAX+ 395 (Luo et al., 2025), Nvidia RTX 5090/B200 (NVIDIA, 2023), etc. Compared with INT4, FP4 is better suited to handling long-tailed distributions (Lee et al., 2024). The use of shared block-scale factors extends the representable dynamic range while simultaneously restricting the influence of outliers.

---

[1]Nanjing University of Science and Technology [2]Institute of Automation, Chinese Academy of Sciences [3]School of Artificial Intelligence, University of Chinese Academy of Sciences [4]Zhongguancun Academy, Beijing, China [5]University of Sydney. Correspondence to: Peisong Wang <peisong.wang@nlpr.ia.ac.cn>, Zhihui Wei <gswei@njust.edu.cn>, Jian Cheng <jcheng@nlpr.ia.ac.cn>.

*Proceedings of the $43^{rd}$ International Conference on Machine Learning*, Seoul, South Korea. PMLR 306, 2026. Copyright 2026 by the author(s).

It supports not only inference but also low-precision training (AMD, 2025; Microsoft, 2024), and can be efficiently emulated or converted on diverse platforms, including Apple M-series chips, NVIDIA Ampere/Ada GPUs, and common x86 CPUs, thus offering broader software and hardware compatibility. To our knowledge, the model openai/gpt-oss (OpenAI, 2025), as the first LLM with native FP4 support, adopts MXFP4, underscoring its importance among future low-precision formats.

Existing W4A4 methods are primarily designed for INT4 quantization and are typically evaluated under different datasets, quantization settings, or simulation modes. As a result, practitioners lack clear guidance on how to apply these methods to the MXFP4 format. To address this gap, we categorize existing PTQ methods into three groups: (1) compensation-based, (2) transformation-based, and (3) optimization-based. We then conduct a detailed comparative analysis of representative methods within each category under the MXFP4 format. Our evaluation highlights methods that achieve significant improvements, and reveals the incompatibility between rotation-based techniques and MXFP4.

Furthermore, we investigate why combining rotation with MXFP4 leads to performance collapse (Lee et al., 2024). To the best of our knowledge, this is the first in-depth study of this issue. We attribute the root cause to a fundamental mismatch: MXFP4 uses a shared PoT (power-of-two) block-scale mechanism to suppress outliers, whereas rotation methods attempt to mitigate them by distributing their energy across all channels. Based on this insight, we propose a grouped rotation strategy to adapt rotation-based methods to MXFP4. This strategy can be easily integrated into existing rotation schemes and substantially improves PTQ accuracy under MXFP4. Our work not only provides practitioners with clear guidance for selecting effective quantization methods but also establishes a direction for further community efforts in optimizing MXFP4 PTQ.

The key contributions of this paper are summarized as follows:

- We conduct a unified empirical evaluation of representative W4A4 PTQ methods under the MXFP4 format, systematically categorizing existing approaches and highlighting their limitations under this emerging format.

- We conduct a thorough investigation of rotation-based methods under the MXFP4 format, identifying that the destructive interaction is fundamentally caused by the combination of PoT scales failing to recover large values within blocks and global rotations amplifying originally small values.

- Building on this insight, we propose a Block-wise Rotation Quantization (BRQ) strategy that adapts rotation

methods to MXFP4. This strategy can be seamlessly integrated into existing rotation schemes and substantially improves PTQ accuracy under MXFP4 across multiple models and tasks.

## 2. Categorization of PTQ Methods

We focus on fully quantized W4A4 PTQ methods, excluding mixed-precision schemes to ensure fair and consistent evaluation under MXFP4. Our evaluation systematically examines existing low-bit PTQ approaches for LLMs in this setting. For clarity, we categorize them into three classes: compensation-based, transformation-based, and optimization-based.

### 2.1. Compensation-Based Quantization Methods

Compensation-based methods reduce quantization errors by adjusting quantized weights to correct low-bit perturbations. GPTQ (Frantar et al., 2022), a representative approach, which performs column-wise offline optimization of weight matrices utilizing second-order information approximations from the Hessian matrix, achieving precise compensation and significantly reducing overall quantization loss. Subsequent methods extend this principle: BoA (Kim et al., 2024) incorporates attention-aware Hessians, RSQ (Sung et al., 2025) applies token-wise weighting, QuantEase (Behdin et al., 2023) leverages coordinate descent for forward reconstruction, VPTQ (Liu et al., 2024a) combines vector quantization with channel-independent second-order optimization, and APTQ (Guan et al., 2024) uses Hessian traces to guide selective mixed-precision quantization.

Together, these compensation-based approaches share the principle of explicit error correction, making them particularly effective for transformer-based LLMs, especially in attention-dense modules sensitive to low-bit perturbations.

### 2.2. Transformation-Based Methods

In the low-bit case, outliers can significantly increase the quantization error. Applying carefully designed equivalent transformations can redistribute or reshape the data to reduce the impact of extreme values. SmoothQuant (Xiao et al., 2023) applies a smoothing transformation to redistribute large activation outliers to the corresponding weight scales, thereby mitigating their impact on low-bit quantization. Building on a similar principle, QServe (Lin et al., 2024b) integrates progressive low-bit quantization with system-level optimization and SmoothAttention to improve inference throughput while maintaining model fidelity. QuIP (Chee et al., 2024) introduces incoherent processing to decorrelate the contributions of outliers in both weight and activation spaces. QuIP# (Tseng et al., 2024) further enhances computational efficiency by employing a random-

*Table 1.* Comparison of WikiText perplexity (Wiki) and average zero-shot accuracy (Avg.) across multiple LLMs under FP16, BINT4, and MXFP4 quantization. QuaRot$^+$ denotes the variant integrated with the GPTQ algorithm. **The best results are highlighted in black bold, while the worst results are highlighted in gray bold.** Detailed results are provided in Appendix .5.

| Method | LLaMA-2 7B | | LLaMA-2 13B | | LLaMA-3 8B | | LLaMA-3.2 1B | | LLaMA-3.2 3B | | Mistral 7B | |
|---|---|---|---|---|---|---|---|---|---|---|---|---|
| | Wiki | Avg. | Wiki | Avg. | Wiki | Avg. | Wiki | Avg. | Wiki | Avg. | Wiki | Avg. |
| FP16 | 5.47 | 62.59 | 4.88 | 64.89 | 6.14 | 65.85 | 9.75 | 53.81 | 7.81 | 61.60 | 5.25 | 66.73 |
| BINT4 | 5.94 | 61.30 | 5.16 | 63.32 | 7.40 | 63.12 | 13.56 | 48.36 | 9.29 | 57.47 | 5.63 | 65.28 |
| RTN | 7.08 | 57.26 | 5.90 | 61.40 | 8.23 | 60.61 | 15.91 | 46.89 | 10.27 | 55.22 | 6.56 | 62.86 |
| GPTQ | 6.56 | **59.27** | 5.41 | 62.91 | 7.68 | 61.48 | 13.35 | 48.52 | **9.50** | 55.40 | 6.00 | 63.34 |
| SmoothQuant | 7.04 | 57.18 | 5.73 | 61.52 | 8.11 | 61.22 | 16.86 | 46.48 | 10.38 | 55.05 | 6.49 | 62.96 |
| QuaRot | *13.09* | *50.32* | *7.03* | *59.09* | *9.56* | *59.26* | *17.86* | *45.42* | *13.36* | *51.60* | *6.65* | *60.33* |
| QuaRot$^+$ | 6.29 | 58.35 | 5.57 | 61.57 | 7.68 | 61.57 | 12.78 | 48.83 | 9.92 | 55.91 | 5.73 | 63.66 |
| OmniQuant | 6.56 | 56.67 | 5.43 | 61.89 | 8.16 | 60.47 | 14.32 | 48.17 | 9.85 | 55.76 | 6.37 | 61.82 |
| SpinQuant | **5.99** | 59.24 | **5.20** | **62.78** | **7.62** | **61.93** | **12.72** | **49.09** | 9.85 | **56.19** | **5.68** | **63.79** |

ized Hadamard transform, which improves orthogonality and reduces inter-channel coherence. QuaRot (Ashkboos et al., 2024) and DuQuant (Lin et al., 2024a) leverage rotation transforms to spread outlier values across subspaces of smaller-magnitude activations or multiple channels, reducing sensitivity to low-bit representation and improving reconstruction accuracy.

These transformation-based methods are particularly effective for modules that exhibit high activation variance or extreme outliers and are fundamental to optimization-based methods.

### 2.3. Optimization-Based Methods

Given the difficulty of manually designing equivalent transformations, some studies propose parameterizing these transformations as learnable variables, allowing them to be optimized within the model to achieve higher performance (Shao et al., 2026). OmniQuant (Shao et al., 2023) introduces learnable weight clipping and equivalent transformations to achieve superior W4A4 quantization performance. SpinQuant (Liu et al., 2024b) demonstrates that optimizing rotation matrices is more effective than random transformations in dispersing weight outliers, significantly reducing quantization errors in extremely low-bit scenarios. AffineQuant (Ma et al., 2024) and FlatQuant (Sun et al., 2024) extend this principle by applying affine transformations to jointly adjust weights and activations, flattening distributions to mitigate the impact of outliers and simplify the optimization process. KurTail (Sadegh Akhondzadeh et al., 2025) leverages kurtosis-based rotation to alleviate outliers in LLM activations, achieving high-fidelity low-bit quantization.

Overall, optimization-driven methods can fully exploit gradient information to adaptively adjust weights and activations under strict low-bit constraints, achieving near-optimal accuracy in low-bit settings.

We discuss recent and concurrent block-level transformation methods in Appendix .12, and clarify our focus on the MXFP4-specific failure mechanism of global rotation.

## 3. Unified Evaluation under MXFP4

To assess whether existing PTQ algorithms fully exploit MXFP4, we conduct a unified empirical evaluation under a consistent MXFP4 setting. Our goal is not to introduce a new reusable benchmark infrastructure, but to provide a controlled comparison that clarifies how representative PTQ methods behave under MXFP4. We test seven state-of-the-art approaches on models of varying scales. To capture overall trends, we average perplexity and downstream accuracy across models and report results in Figure 1. This unified comparison provides a fair basis for analysis and reveals key limitations that motivate the proposed method.

### 3.1. Experimental Setup

All experiments are carried out on NVIDIA A800 GPU servers, with MX format quantization simulated using Microsoft's open-source repository *microsoft/microxcaling* (Microsoft, 2024).

**Method selection.** We selected representative PTQ methods from the three categories in Section 2. For compensation-based methods, we chose GPTQ (Frantar et al., 2022). Transformation-based methods include SmoothQuant (Xiao et al., 2023) and QuaRot (Ashkboos et al., 2024). Notably, we distinguish between two variants of QuaRot: QuaRot, which applies random rotation with RTN, and QuaRot+, which integrates random rotation with GPTQ, in order to separately evaluate the effect of random rotation alone and in combination with GPTQ. Optimization-based methods comprise OmniQuant (Shao et al., 2023) and SpinQuant (Liu et al., 2024b), representing parameter optimization and end-to-end rotation optimization, respectively. We also include round-to-nearest (RTN) INT4 with block size 32 and FP16

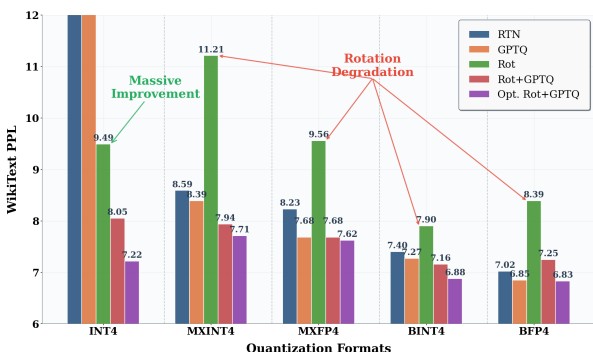

*Figure 2.* Effect of rotation and its variants across different quantization formats. Rot applies a random Hadamard transform with RTN; Rot+GPTQ combines the transform with GPTQ; and Opt. Rot+GPTQ employs an optimized rotation matrix with GPTQ.

scale as a naive baseline (BINT4).

**Models.** We benchmarked selected methods on multiple widely adopted large language models, including LLaMA-2 7B/13B, LLaMA-3 8B, LLaMA-3.2 1B/3B and Mistral-7B, which span different scales and architectures. These models represent a spectrum of modern transformer-based LLMs and provide a robust testbed for quantization research.

**Datasets and metrics**. We evaluated perplexity (PPL) on WikiText2 as a proxy for language modeling quality, and accuracy on five zero-shot downstream tasks: PIQA (Bisk et al., 2020), WinoGrande (Sakaguchi et al., 2021), Open-BookQA (Mihaylov et al., 2018), ARC-Easy and ARC-Challenge (Boratko et al., 2018).

### 3.2. Evaluations

As shown in Table 1, MXFP4 RTN suffers a substantial accuracy drop compared to FP16 and even BINT4 baselines. This demonstrates that despite MXFP4's significant hardware advantages, PTQ on it remains a significant challenge, further highlighting the need to systematically evaluate the performance of existing PTQ methods on MXFP4.

In addition, we can see that most methods yield some improvements when directly applied to the MXFP4 format. GPTQ stands out by consistently delivering notable gains, even surpassing BINT4 on certain models (e.g., LLaMA-3.2 1B: 15.91/46.89 → 13.35/48.52). However, other methods are less reliable. OmniQuant requires delicate hyperparameter tuning to achieve stable optimization on small models (LLaMA-3.2 1B/3B), yet still underperforms GPTQ (e.g., LLaMA 3.2 1B PPL 14.32 vs. 13.35). SmoothQuant provides only marginal benefits and can even harm performance, revealing MXFP4's heightened sensitivity to parameter magnitudes under its scaling scheme.

The most striking results arise from rotation-based methods. QuaRot, when combined with RTN, leads to catastrophic degradation (e.g., LLaMA-2 7B: 7.08/57.26 → 13.09/50.32).

Even when integrated with GPTQ, performance gains remain inconsistent and limited. This indicates a structural incompatibility between random rotations and MXFP4, but the root cause of this destructive interaction has not yet been thoroughly discussed by research (Lee et al., 2024). SpinQuant leverages a straight-through estimator for end-to-end optimization, which enables rotations to adaptively align with the non-uniform scaling of MXFP4. While this enforced optimization does alleviate the incompatibility to some extent, it delivers only marginal improvements over QuaRot$^+$ (e.g., Mistral 7B: 5.73/63.66 → 5.68/63.79), suggesting that optimization alone does not fully resolve the compatibility issue.

Given the critical role of rotation in INT4 quantization, we further examine its impact across other commonly used quantization formats. Figure 2 reports results for rotation and its variants under INT4 (weight per-channel symmetric quantization with per-token asymmetric activation quantization), BINT4, BFP4 (FP4 variant of BINT4), as well as MXINT4 and MXFP4. The key findings can be summarized as follows:

**A. INT4 benefits substantially from rotation.** In the widely studied INT4 setting, applying rotation alone yields significant performance improvements. When combined with GPTQ or rotation optimization, the gains are further amplified, indicating that rotation is particularly effective for uniformly distributed integer formats.

**B. FP4 formats outperform INT4 without rotation.** When not using rotation, BFP4 and MXFP4 achieve consistently higher performance than BINT4 and MXINT4, suggesting that FP4's wider dynamic range and representational flexibility are better suited for 4-bit quantization.

**C. Random rotation degrades performance in group-wise formats.** In contrast to INT4, group-wise quantization formats (BINT4, BFP4, MXINT4, MXFP4) suffer from performance degradation under random rotation. The effect is especially pronounced in MX-based formats, where performance can drop below that of simple RTN.

**D. Divergent behaviors under FP16 vs. PoT scaling.** For FP16-scale formats, BINT4 outperforms BFP4 after random rotation. Conversely, for PoT-scale formats, MXFP4 underperforms compared to its INT4 counterpart (MXINT4).

**E. PoT scaling in MX formats incurs additional loss.** Comparing MXINT4/MXFP4 against their FP16-scale counterparts BINT4/BFP4, PoT scaling consistently introduces larger quantization errors, which become even more severe after rotation.

**F. Optimized rotation remains limited on MXFP4.** While optimized rotation combined with GPTQ improves MXFP4 performance, the final results still lag behind those of INT4

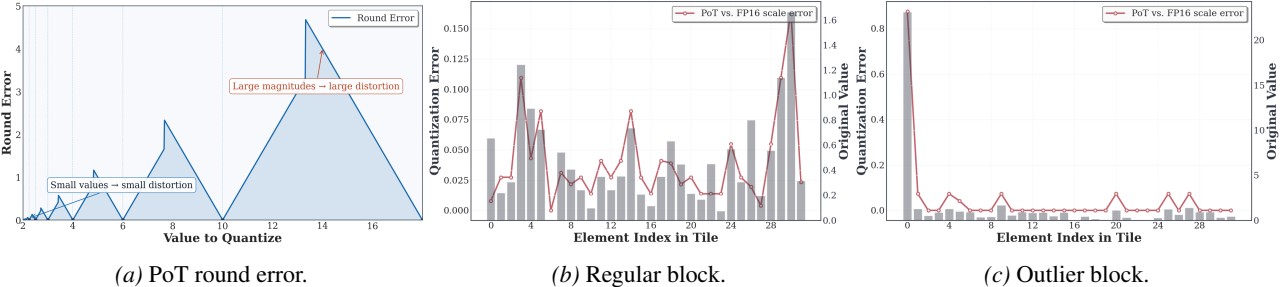

*(a)* PoT round error.        *(b)* Regular block.        *(c)* Outlier block.

*Figure 3.* (a) illustrates the rounding error curve of PoT format. (b) and (c) show the quantization error of MXFP4 relative to BFP4 for regular and outlier blocks, respectively. Bar charts represent the original activation values (right axis), lines indicate the relative quantization error (left axis).

under comparable configurations.

Overall, these results reveal a striking divergence: while rotation and its variants consistently enhance INT4, they fail to generalize to MXFP4. In particular, the interaction between rotation and MXFP4's block-wise PoT scaling leads to unique degradation patterns. This raises an important open question: **Why does a technique that is fundamentally beneficial in INT4 become harmful in MXFP4?** To address this, we next conduct a deeper analysis of MXFP4's structural characteristics and their destructive interplay with rotation-based transformations.

## 4. Why Rotation Transforms Hurt MXFP4

To understand why rotation transformations—despite their remarkable success on INT4—degrade quantization accuracy under the MXFP4 format, we first dissect the unique characteristics and inherent limitations of MXFP4. We then analyze how rotation reshapes model data distributions, and by synthesizing these perspectives, we identify the root cause of their conflict. Building on this insight, we further propose a practical solution to reconcile the incompatibility.

### 4.1. Limited Recovery of Large Values in MXFP4 Blocks

MXFP4 represents each value in the E2M1 format, with one sign bit, one mantissa bit, and two exponent bits. It applies symmetric per-group quantization with a fixed group size of 32, where each group is associated with an E8M0 (PoT) scaling factor directly integrated into hardware. With finer granularity and FP4's non-uniform representation, MXFP4 achieves a substantially higher quantization signal-to-noise ratio (QSNR) than per-tensor or per-channel INT4, thus better approximating full-precision values (Darvish Rouhani et al., 2023).

Figure 2 shows that BFP4 and MXFP4 can lead to significant performance differences due to different scale formats. To further investigate the differences between these scales, we categorize quantization blocks into two types: "**regular blocks**," which contain no outliers, and "**outlier blocks**,"

which include one or more outliers (here outliers are defined as the top 0.1% of activations in descending order of absolute value (Dettmers et al., 2022)). Figure 3b and 3c visualize MXFP4's quantization error relative to BFP4 for both block types. We observe that for both regular and outlier blocks, the error increases with the magnitude of the elements. Notably, in outlier blocks, the quantization loss of outliers is up to five times larger than the maximum loss in regular blocks. This is primarily due to the PoT format's coarse granularity at large magnitudes, which amplifies rounding errors when representing large values (see Figure 3a).

In summary, the main bottleneck of MXFP4 lies in its limited ability to reconstruct large values in blocks, with the reconstruction error increasing sharply with magnitude. Therefore, improving MXFP4 performance ultimately depends on effectively reducing these large values.

### 4.2. Rotation Induced Growth of Small Values

In W4A4 quantization, the primary source of performance degradation is the quantization error of activations (Ashkboos et al., 2024). To investigate the compatibility issues between rotation-based transformations and the MXFP4 format, we conducted a detailed analysis of activation distributions before and after rotation.

As illustrated in Figure 4, conventional rotation methods employ rotation matrices to redistribute outliers originally concentrated in a few channels across all dimensions, thereby reducing quantization error. However, rotation does not reduce the overall energy; the L2 norm of the activations remains unchanged. In effect, the energy from the original outlier channels is not eliminated but redistributed across previously small-value channels, which consequently become magnified. To examine this effect, we sampled 2,048 activations from LLaMA-3 8B and analyzed their distributional shifts after rotation, as shown in Figure 5. The results indicate that rotation largely removes the ∼1% of activations exceeding 3 (corresponding to the blue area in the figure), but at the cost of substantially **increasing the proportion of activations greater than 1.5** (from about 5%

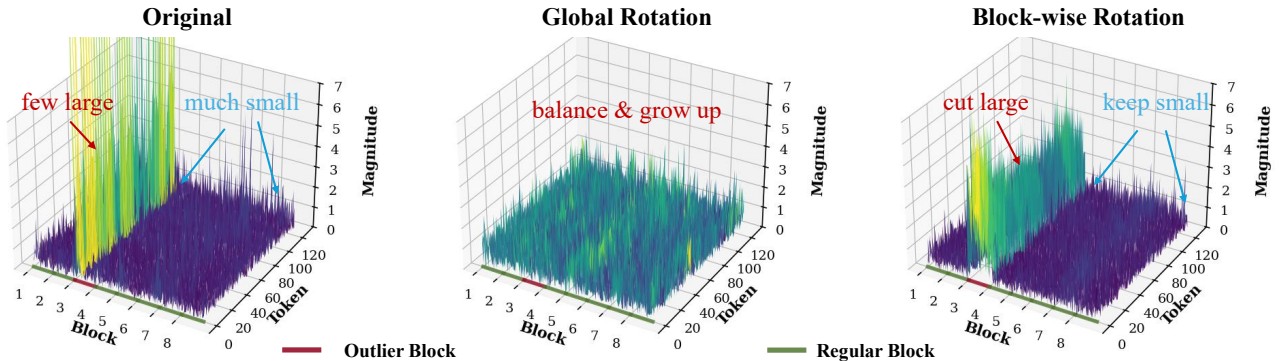

*Figure 4.* Comparison of the distribution of Llama-3 8B activation after different transformations. More block-scale visualizations are provided in **Appendix .4**.

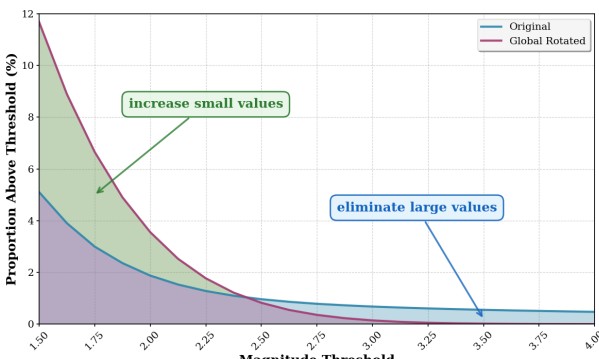

*Figure 5.* The effect of rotation transformation on activation distribution. The horizontal axis represents the segmentation threshold, and the vertical axis represents the percentage of data greater than the threshold.

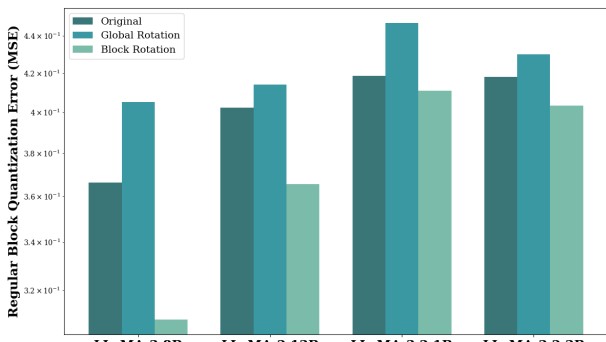

*Figure 6.* Average quantization loss (logarithmic result) of regular blocks after applying different rotations, where outliers are defined as the top 0.1% of activations in descending order of absolute value (Dettmers et al., 2022).

before rotation to 11% after rotation, corresponding to the green area in the figure). This evidence clearly demonstrates the effect of rotation on **amplifying small-value blocks**.

Synthesizing the above observations, we attribute the incompatibility between rotation and MXFP4 quantization to the following destructive interactions:

- Global rotation amplifies the scales of regular blocks, thereby increasing their quantization difficulty.

- The poor reconstruction of large values within MXFP4 blocks further exacerbates the quantization error of these amplified regular blocks.

- Since regular blocks vastly outnumber outlier blocks, the accumulated errors across them dominate, ultimately leading to a substantial increase in overall quantization loss after rotation.

To validate this inference, we measure the average quantization error of regular blocks across different models before and after applying global rotation. As shown in Figure 6, regular-blocks' quantization losses significantly increase after rotation, providing strong evidence for our conjecture. Since regular blocks vastly outnumber outlier blocks, this

imbalance ultimately leads to the collapse of quantization accuracy under MXFP4 when global rotation is applied.

### 4.3. Theoretical Analysis

**Analysis 1: Rotation increases the scale of regular blocks.**

Let $X = [x_1, x_2, \ldots, x_D] \in \mathbb{R}^D$ be an activation vector, and let $H \in \mathbb{R}^{D \times D}$ denote a Hadamard transform. Define the activation "energy" as the squared $\ell_2$ norm, $\mathbb{E}(X) = \|X\|_2^2$. Since orthogonal rotations preserve the $\ell_2$ norm, we have $\mathbb{E}(HX) = \|HX\|_2^2 = \|X\|_2^2 = \mathbb{E}(X)$. Let $\Omega$ be the index set of outlier channels in $X$, with the remaining indices regarded as regular channels. Suppose there exist thresholds $\alpha$ and $\beta$ such that for all $i \notin \Omega$, $|x_i| \leq \alpha$, and for all $i \in \Omega$, $|x_i| \geq \beta$, with $\beta \gg \alpha$. Consider block-wise quantization with block size $B$. Let $s_i$ be the quantization scale for block $i$, $i = 1, 2, \ldots, D/B$. Let $\Omega_b$ be the index set of outlier blocks (i.e., blocks containing at least one outlier channel). Then $s_i \geq \beta$ for $i \in \Omega_b$ and $s_i \leq \alpha$ for $i \notin \Omega_b$. Assume an idealized setting in which the rotation approximately equalizes energy across channels. Then each block approximately receives an equal share of the total

energy. For a regular block $X_i$ ($i \notin \Omega_b$), the pre-rotation energy is bounded by $\mathbb{E}_{\mathrm{br}}(X_i) \leq B\alpha^2$. After rotation, the block energy is approximately

$$\mathbb{E}_{\mathrm{ar}}(X_i) \approx \frac{\mathbb{E}(X)}{D/B} \geq \frac{nB\beta^2}{D}, \tag{1}$$

where $n = |\Omega|$ is the number of outlier channels. Therefore,

$$\frac{\mathbb{E}_{\mathrm{ar}}(X_i)}{\mathbb{E}_{\mathrm{br}}(X_i)} \geq \frac{n\beta^2}{D\alpha^2}. \tag{2}$$

Here, $n/D$ is the fraction of outlier channels, and $\beta/\alpha$ is the outlier-to-regular magnitude ratio. Prior work typically reports $n/D$ in the 0.1%–1% range and $\beta/\alpha$ in the hundreds to thousands (Raman et al., 2025; An et al., 2025; Xiang & Zhang, 2024). This implies that, after rotation, the energy of regular blocks is larger than before, and thus their scales increase. This matches the visual patterns shown in Figure 4.

**Analysis 2: Larger scale increases quantization error.**

Let $X$ be an activation to be quantized with symmetric min–max scaling. Let $s$ denote the scale, equivalently mapping the range $[-a, a]$ (with $a = \max |X|$) to the quantization grid. Consider uniform integer quantization with bit-width $b$, giving $L = 2^b - 1$ grid points. The quantization step size is

$$\Delta = \frac{2a}{L-1} \approx \frac{2a}{L}. \tag{3}$$

Let $Q(\cdot)$ denote rounding-and-clipping, and let the error be $\mathcal{E} = X - Q(X)$. Within a single quantization bin of width $\Delta$, the error is approximately distributed symmetrically in $[-\Delta/2, \Delta/2]$. The per-bin mean squared error (MSE) is thus the variance of a uniform distribution,

$$\mathrm{MSE}_{\mathrm{bin}} \approx \mathbb{E}[e^2] = \frac{\Delta^2}{12}. \tag{4}$$

Hence the overall MSE scales proportionally to $\Delta^2$. Since $\Delta \propto a$ (or equivalently to the effective scale), we get

$$\mathrm{MSE} \propto \Delta^2 \propto a^2. \tag{5}$$

In other words, when min–max scaling increases the scale (or the maximum absolute value $a$), the quantization error grows quadratically. Although MXFP4 uses a floating-point-like code, its "scale" (i.e., the exponent offset/dynamic range that aligns the block) determines the spacing between representable values, which typically expands with scale (linearly or exponentially depending on the exponent). Whether fixed-point or floating-point quantization, enlarging the representable spacing increases the error bound; in practice the MSE growth remains well-approximated as scaling with the square of the effective scale. Therefore, the conclusion holds: **larger scale → larger spacing → larger quantization error.**

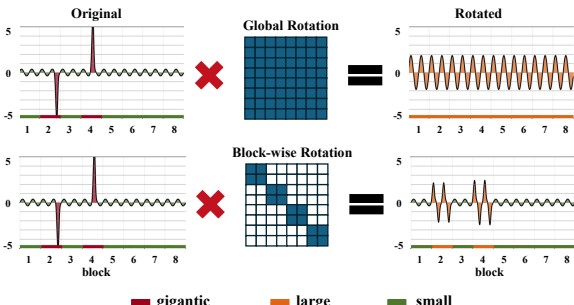

*Figure 7.* Block rotation's intuition: Global rotations spread outliers across all channels, inflating regular block scales and worsening quantization error. Block-wise rotations redistribute outliers locally, mitigating outlier effect while keeping regular block scales intact, thereby minimizing quantization error.

In addition, MXFP4 also has rounding error due to scale quantization. Since the PoT step size grows exponentially, the expected rounding error will also change exponentially with the increase of scale, as shown in Figure 3a, thereby further increasing the overall quantization error.

Putting **Analysis 1** and **Analysis 2** together, global rotation increases the quantization error of regular blocks, consistent with the statistics in Figure 6.

### 4.4. Fix Rotation in MXFP4

To mitigate the incompatibility between rotation and the block-wise PoT scale inherent to MXFP4, we propose a block-wise rotation strategy, which applies rotation transformations independently within each quantization block, as shown in Figure 7. Unlike global rotation, which mixes outliers across all channels, block-wise rotation partitions activations into fixed-size groups (aligned with MXFP4 blocks, e.g., 32 channels) and performs an independent orthogonal transformation within each group. This design preserves the denoising effect of rotation while preventing excessive amplification of small-value channels caused by global mixing. The rotation matrix fusion method and position are consistent with SpinQuant.

Formally, let $x \in \mathbb{R}^N$ denote the activation vector for the linear, partitioned into $B$ blocks of size $g$ ($N = B \times g$). Block-wise rotation constructs a block-diagonal matrix:

$$R_{\mathrm{block}} = \mathrm{diag}(R_1, R_2, \ldots, R_B), \quad R_i \in \mathbb{R}^{g \times g}, \; R_i^\top R_i = I. \tag{6}$$

Block-wise rotation offers multiple advantages over global rotation:

- **Outlier suppression:** By applying rotations independently within each block, the block rotation matrix effectively redistributes outliers that were previously concentrated in a few channels. This discretization prevents any

single outlier from dominating the quantization process, thereby reducing the quantization error of outlier blocks.

- **Controlled quantization error:** Because each block is rotated independently, the amplification of small-value channels caused by rotation is confined within the block itself. This prevents error propagation across blocks, keeps the quantization loss in regular blocks under control, which is a key issue in global rotation.

- **Reduced online computing:** Block-wise rotation substantially reduces the online rotation computation (same as SpinQuant's $R_4$) before the $down_{proj}$ layer. For an input dimension of $N$, global rotation incurs $O(N^2)$ complexity, whereas block-wise rotation reduces it to $O(N \times 32)$. This reduction not only enhances computational efficiency but also facilitates the practical deployment of rotation-based quantization at scale.

## 5. Experiment

In this section, we evaluate BRQ using the same test sets and hyperparameters as in Section 3. Comparisons are made with GPTQ, QuaRot$^+$, SpinQuant, RTN, and BINT4. Beyond the LLaMA and Mistral models, we also include evaluations on the Qwen (Team, 2024) model.

### 5.1. Main Results

Table 2 presents the results of combining block-wise randomized Hadamard rotations with GPTQ under the MXFP4 format. Compared to the global rotation in QuaRot$^+$, BRQ delivers substantial improvements, surpassing the strong BINT4 baseline on most models. Particularly, on the more challenging LLaMA-3.2 1B and Qwen2.5 1.5B models, BRQ reduces perplexity from 12.78/12.80 to 11.95/12.15 and raises downstream task accuracy from 48.83/53.50 to 49.87/54.83. These results confirm that block-wise rotation is key to reconciling rotation-based methods with MXFP4, further corroborating our analysis in Section 4.

We next evaluate the benefits of optimizing block rotation matrices. Table 3 compares SpinQuant, BRQ with randomized block rotations, and BRQ$_{Spin}$, where block rotations are optimized within the SpinQuant framework. Following the SpinQuant setup, we adopt Cayley SGD (Li et al., 2020) and optimize using 800 sequences of length 2048 from Wikitext2.

Even without optimization, BRQ already outperforms Spin-Quant with optimized global rotations. For instance, on LLaMA-3 8B, BRQ reduces perplexity from 7.62 (Spin-Quant) to 7.14, also surpassing BINT4 (7.40). On downstream tasks, accuracy improves from 61.93 (SpinQuant) to 63.54, narrowing the error gap by 41%. Similar trends hold across other models. Given that SpinQuant incurs high opti-

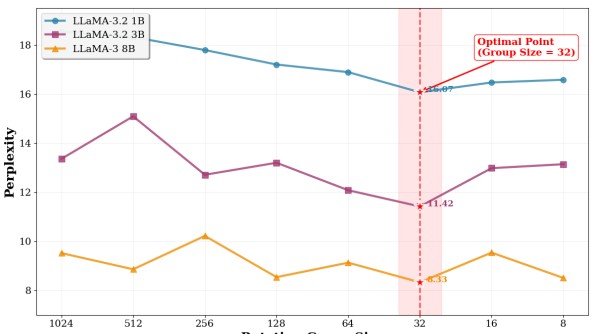

*Figure 8.* Impact of rotation group size on MXFP4 quantization performance. Across different LLaMA models, the lowest perplexity is consistently obtained when the rotation group size is aligned with the MXFP4 block size of 32.

mization costs (Liu et al., 2024b), BRQ's ability to achieve better performance without optimization undoubtedly enhances its practicality for real-world deployment.

Moreover, optimization brings further gains: for LLaMA-3.2 3B, perplexity decreases from 9.41 with random block rotations to 9.08 after optimization, significantly better than SpinQuant's 9.85. However, the improvements remain limited for many models, such as LLaMA-3 8B and Mistral 7B. These findings not only demonstrate the compatibility of BRQ with existing frameworks but also suggest the potential limitations of the SpinQuant optimization scheme.

### 5.2. Effect of Rotation Dimension

To further verify the fit of rotation dimension size to MXFP4, Figure 8 reports the PPL results of LLaMA-3 8B and LLaMA-3.2 1B/3B with different rotation dimensions. We observe that using larger rotation dimensions amplifies the impact of outliers and thus increases the overall loss, while smaller dimensions suffer from insufficient discrete channels. The best PPL is achieved when the rotation dimension matches the MXFP4 block size, which is consistent with our analysis in Section 4.

### 5.3. Performance Analysis

**Simulation experiments on NVIDIA A800.** We implement our method on PyTorch with CUDA 12.4 and employ *microsoft/microxcaling* (Microsoft, 2024) for MXFP4 quantization. In this section, we compare the performance of BRQ and QuaRot on both the prefill and decode stages using NVIDIA A800 GPUs.

We evaluate the prefill speed of LLaMA-2 7B models, with results reported in Appendix .8 Table 19 (generation in Table 20). As expected, the block-wise rotation in BRQ drastically reduces the online rotation computation. Compared to QuaRot, BRQ lowers the additional inference latency caused by rotation by 40%, further improving the practicality of rotation-based quantization methods. Note that

*Table 2.* Performance comparison of BRQ using randomized block rotations and existing PTQ methods without optimization. LLaMA-2 7B/13B/70B, Qwen2.5 7B and Mixtral 8×7B results can be found in the Appendix .3/ .5.

| Method | LLaMA 3 8B | | LLaMA 3.2 1B | | LLaMA 3.2 3B | | Mistral 7B | | Qwen2.5 1.5B | | Qwen2.5 3B | |
| --- | --- | --- | --- | --- | --- | --- | --- | --- | --- | --- | --- | --- |
| | Wiki | Avg. | Wiki | Avg. | Wiki | Avg. | Wiki | Avg. | Wiki | Avg. | Wiki | Avg. |
| FP16 | 6.14 | 65.85 | 9.75 | 53.81 | 7.81 | 61.60 | 5.25 | 66.73 | 9.87 | 58.83 | 8.03 | 61.91 |
| BINT4 | 7.40 | 63.12 | 13.56 | 48.36 | 9.29 | 57.47 | 5.63 | 65.28 | 13.98 | 53.98 | 10.32 | 58.03 |
| RTN | 8.23 | 60.61 | 15.91 | 46.89 | 10.27 | 55.22 | 6.56 | 62.86 | 16.61 | 52.69 | 11.03 | 57.77 |
| GPTQ | 7.68 | 61.48 | 13.35 | 48.52 | 9.50 | 55.40 | 6.00 | 63.34 | 13.94 | 53.13 | 10.20 | 58.23 |
| QuaRot$^+$ | 7.68 | 61.57 | 12.78 | 48.83 | 9.92 | 55.91 | 5.73 | 63.66 | 12.80 | 53.50 | 9.65 | 58.27 |
| BRQ | **7.14** | **63.54** | **11.95** | **49.87** | **9.41** | **56.88** | **5.59** | **64.19** | **12.15** | **54.83** | **9.48** | **59.68** |

*Table 3.* Performance comparison of optimized block rotation transformation (BRQ$_{Spin}$), random block rotation transformation (BRQ), and optimized global rotation transformation (SpinQuant).

| Method | LLaMA 3 8B | | LLaMA 3.2 1B | | LLaMA 3.2 3B | | Mistral 7B | | Qwen2.5 1.5B | | Qwen2.5 3B | |
| --- | --- | --- | --- | --- | --- | --- | --- | --- | --- | --- | --- | --- |
| | Wiki | Avg. | Wiki | Avg. | Wiki | Avg. | Wiki | Avg. | Wiki | Avg. | Wiki | Avg. |
| FP16 | 6.14 | 65.85 | 9.75 | 53.81 | 7.81 | 61.60 | 5.25 | 66.73 | 9.87 | 58.83 | 8.03 | 61.91 |
| SpinQuant | 7.62 | 61.93 | 12.72 | 49.09 | 9.85 | 56.19 | 5.68 | 63.79 | 12.64 | 53.57 | 9.58 | 59.09 |
| BRQ | 7.14 | **63.54** | 11.95 | 49.87 | 9.41 | 56.88 | 5.59 | 64.19 | 12.15 | 54.83 | 9.48 | **59.68** |
| BRQ$_{Spin}$ | **7.13** | 63.39 | **11.93** | **50.00** | **9.08** | **57.29** | **5.57** | **64.26** | **11.95** | **55.07** | **9.46** | 59.65 |

*Table 4.* End-to-end prefill latency comparison on LLaMA-3.2 1B with NVIDIA RTX 5090 under MXFP4 execution.

| Method | SeqLen 256 | | SeqLen 512 | |
| --- | --- | --- | --- | --- |
| | Latency (ms) | Overhead | Latency (ms) | Overhead |
| BF16 | 100.60 | – | 201.77 | – |
| MXFP4 | 53.25 | – | 89.24 | – |
| QuaRot | 55.20 | 3.66% | 95.85 | 7.41% |
| BRQ | 54.03 | 1.46% | 89.91 | 0.75% |

since the A800 does not natively support FP4 inference, inference speed would be further improved on dedicated MXFP4 hardware.

**Hardware measurements on NVIDIA RTX 5090.** To further evaluate the practical efficiency of BRQ on hardware with MXFP4 execution support, we conduct end-to-end prefill latency measurements on LLaMA-3.2 1B using NVIDIA RTX 5090. Different from the A800 experiments above, this evaluation is performed under a hardware-facing setting and is intended to validate whether the latency advantage of block-wise rotation can also be observed in practice.

As shown in Table 4, BRQ consistently achieves lower latency than global rotation-based QuaRot. At sequence lengths 256 and 512, BRQ reduces the prefill latency of QuaRot from 55.20 ms to 54.03 ms and from 95.85 ms to 89.91 ms, corresponding to relative latency reductions of 2.1% and 6.2%, respectively. More importantly, the extra overhead over native MXFP4 is reduced from 3.66%/7.41% to 1.46%/0.75%. These results show that BRQ remains close to native MXFP4 efficiency while avoiding the larger runtime overhead introduced by global rotation.

## 6. Conclusion

In this study, we systematically evaluated representative INT4-oriented PTQ algorithms under the MXFP4 format, systematically revealing the limitations of existing methods in this hardware-friendly setting. We analyzed the incompatibility between MXFP4 and rotation-based approaches, identifying the conflict between block-wise quantization and rotation-induced energy redistribution, amplified by MXFP4's PoT scale. To address this, we introduce BRQ, which adapts rotation to MXFP4, resolving compatibility issues and significantly improving PTQ performance.

This work aims to provide both theoretical insights and practical guidance for deploying large language models on next-generation low-bit floating-point hardware.

## Acknowledgements

This work was supported by the National Natural Science Foundation of China (Grant No. 62572471), the Natural Science Foundation of Jiangsu Province (Grant No. BK20243051), and the Science and Technology Major Special Program of Jiangsu (Grants No. BG2024028).

## Impact Statement

This paper studies efficient PTQ for LLMs under low-bit floating-point formats. It may reduce inference cost, memory use, and energy consumption. The method does not introduce new model capabilities, use personal data, or change training objectives. We foresee no direct negative impact.

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

# Appendix

## .1. Use of LLMs

The text in this paper has been professionally refined using LLM to enhance clarity, coherence, and adherence to academic writing standards.

## .2. Experimental Details

In this section, we provide the detailed hyperparameter settings used for each method in the benchmark, to ensure reproducibility of our experiments. All methods employ WikiText2 as the calibration dataset, and were conducted on NVIDIA A800 GPUs.

- **FP16:** All models are configured to use float16 for FP16 benchmarking.

- **GPTQ:** Calibrated with 128 sequences of length 2048. The damping parameter for Hessian estimation is set to 0.01, following the authors' recommendation (Frantar et al., 2022).

- **SmoothQuant:** Calibrated with 512 sequences of length 512 for activation scaling. The smoothing coefficient $\alpha$ is set to 0.85, consistent with the official repository (Xiao et al., 2023).

- **OmniQuant:** Calibrated with 128 sequences of length 2048, with optimization applied to learnable weight clipping and equivalent transformations (Shao et al., 2023).

- **SpinQuant:** Calibrated with 800 sequences of length 2048, optimized via the Cayley SGD (Li et al., 2020) optimizer for rotation matrix training (Liu et al., 2024b).

- **BRQ:** BRQ follows SpinQuant, but replaces global rotation with block-wise rotation. The block-wise rotations are fused/appended at the same locations as in SpinQuant (including R1, R2 and R4), which consists of 32-dimensional random Hadamard transformations, as shown in Equation 6. The paper uses randomized block Hadamard transform by default for BRQ, while BRQ$_{Spin}$ uses the SpinQuant-optimized block Hadamard transform. After fusing the rotations, we quantize weights to 4 bit with GPTQ (MXFP4).

## .3. Application on 70B model

To explore the effect of BRQ on larger-scale models, we conducted comparative experiments on LLaMA-2 70B. As shown in Table 5, BRQ consistently outperforms existing methods at this scale. In particular, compared with QuaRot—which also leverages randomized Hadamard transforms—BRQ reduces the perplexity of LLaMA-2 70B from 3.76 to 3.62, while improving the average downstream accuracy from 68.86 to 69.10. Due to memory limitations on our available servers, we were unable to apply SpinQuant to the 70B model, and thus its results are not reported here. These results further demonstrate the effectiveness of BRQ on large-scale models.

*Table 5.* Performance of different quantization methods on LLaMA-2 70B.

| Method | Wiki | WG | PIQA | OBQA | ARC-E | ARC-C | Avg. |
|---|---|---|---|---|---|---|---|
| FP16 | 3.32 | 77.97 | 82.75 | 48.80 | 81.01 | 57.50 | 69.61 |
| RTN | 4.20 | 75.92 | 80.90 | 46.00 | 78.53 | 53.32 | 66.93 |
| GPTQ | 3.79 | 75.37 | 80.84 | 48.00 | 79.92 | 54.60 | 67.75 |
| QuaRot$^{+}$ | 3.76 | 77.66 | 81.82 | 47.80 | 79.75 | 57.25 | 68.86 |
| BRQ | 3.62 | 77.68 | 82.54 | 47.60 | 80.30 | 57.40 | 69.10 |

## .4. Rotation Comparison

Figure 9 illustrates the block-wise scale distribution (defined as the maximum absolute value within each block) under the original activations and after applying global rotation. In the original activations, only a small fraction of blocks contain outliers, while most blocks maintain relatively small scales. After global rotation, although the prominent outliers are

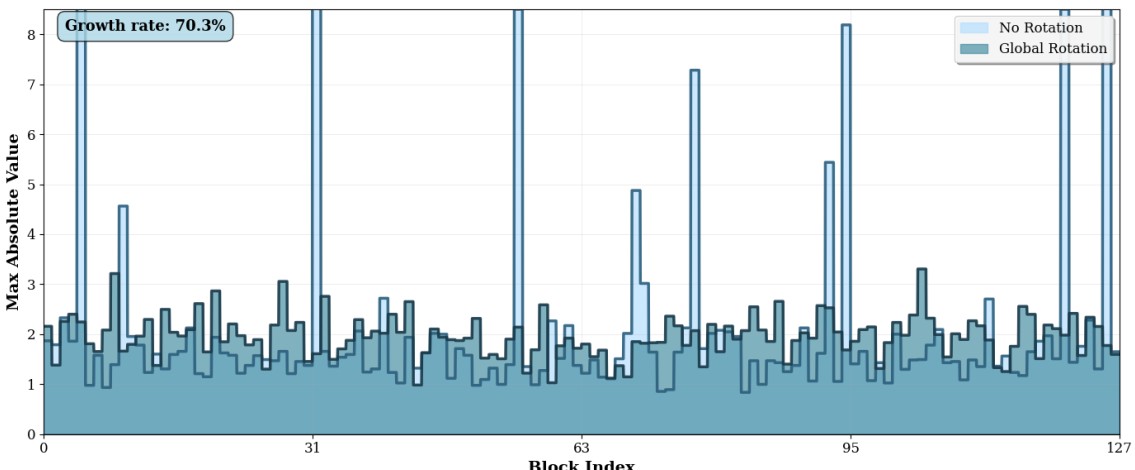

*Figure 9.* Changes in block maximum values after applying global rotation.

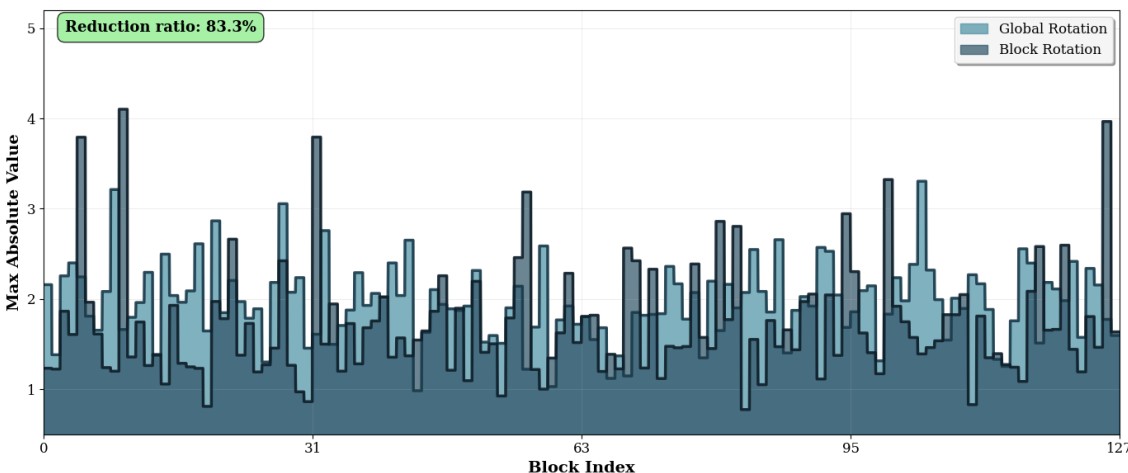

*Figure 10.* Comparison of group maximum values after global rotation and block rotation.

mitigated, the scales of more than 70% of the regular blocks increase substantially. This amplification of regular-block scales is the fundamental reason behind the collapse of quantization performance under MXFP4 following global rotation.

We further analyze the blocks that exhibited significant scale growth in Figure 10 and evaluate the impact of block-wise rotation on these blocks. Figure 10 compares the scale distributions under global rotation and block-wise rotation. We observe that block-wise rotation mitigates over 80% of the scale inflation introduced by global rotation. This result provides strong evidence that block-wise rotation effectively alleviates the quantization degradation caused by global rotation, thereby confirming its effectiveness.

## .5. Detailed Results

This section presents the detailed results of the experiments reported in the main text. Notably, in these experiments, BRQ employs block-wise stochastic Hadamard matrices for rotation without any additional optimization of the rotation matrices.

### .5.1. LLaMA Results

The following are the detailed experimental results of the LLaMA family series models in Tables 1 and 2.

*Table 6.* Evaluation of different methods on LLaMA-2 7B across multiple benchmarks.

| Method | Wiki | WG | PIQA | OBQA | ARC-E | ARC-C | Avg. |
|--------|------|------|------|------|-------|-------|------|
| FP16 | 5.47 | 68.98 | 79.05 | 44.20 | 74.57 | 46.16 | 62.59 |
| BINT4 | 5.94 | 68.19 | 76.93 | 43.00 | 73.65 | 44.71 | 61.30 |
| RTN | 7.08 | 64.80 | 76.39 | 39.20 | 65.70 | 40.19 | 57.26 |
| SmoothQuant | 7.04 | 64.64 | 76.17 | 39.00 | 66.75 | 39.33 | 57.18 |
| GPTQ | 6.56 | 66.61 | 76.55 | 40.60 | 71.12 | 41.46 | 59.27 |
| OmniQuant | 6.56 | 63.06 | 76.33 | 36.60 | 67.09 | 40.27 | 56.67 |
| QuaRot | 13.09 | 59.11 | 71.21 | 34.40 | 55.05 | 31.82 | 50.32 |
| QuaRot$^+$ | 6.29 | 67.24 | 75.68 | 39.80 | 69.02 | 40.01 | 58.35 |
| SpinQuant | 5.99 | 66.14 | 77.69 | 40.40 | 70.58 | 41.38 | 59.24 |
| BRQ | 5.84 | 67.09 | 76.77 | 44.80 | 73.23 | 43.17 | 61.01 |

*Table 7.* Evaluation of different methods on LLaMA-3 8B across multiple benchmarks.

| Method | Wiki | WG | PIQA | OBQA | ARC-E | ARC-C | Avg. |
|--------|------|------|------|------|-------|-------|------|
| FP16 | 6.14 | 73.16 | 80.57 | 44.80 | 77.56 | 53.15 | 65.85 |
| BINT4 | 7.40 | 70.40 | 78.94 | 43.60 | 73.95 | 48.72 | 63.12 |
| RTN | 8.23 | 67.56 | 77.31 | 41.80 | 70.74 | 45.64 | 60.61 |
| SmoothQuant | 8.11 | 67.88 | 78.29 | 43.60 | 71.00 | 45.31 | 61.22 |
| GPTQ | 7.68 | 70.48 | 76.06 | 42.00 | 73.23 | 45.64 | 61.48 |
| OmniQuant | 8.16 | 66.06 | 77.20 | 40.60 | 72.85 | 45.65 | 60.47 |
| QuaRot | 9.56 | 67.71 | 75.57 | 41.60 | 69.02 | 42.40 | 59.26 |
| QuaRot$^+$ | 7.68 | 68.16 | 75.36 | 42.80 | 72.66 | 48.89 | 61.57 |
| SpinQuant | 7.62 | 69.56 | 76.93 | 42.00 | 72.90 | 48.25 | 61.93 |
| BRQ | 7.14 | 71.98 | 78.51 | 42.60 | 75.04 | 49.57 | 63.54 |

*Table 8.* Evaluation of different methods on LLaMA-2 13B across multiple benchmarks.

| Method | Wiki | WG | PIQA | OBQA | ARC-E | ARC-C | Avg. |
|--------|------|------|------|------|-------|-------|------|
| FP16 | 4.88 | 72.13 | 80.52 | 45.20 | 77.44 | 49.14 | 64.89 |
| BINT4 | 5.16 | 72.06 | 78.84 | 43.20 | 75.55 | 46.93 | 63.32 |
| RTN | 5.90 | 69.45 | 77.25 | 42.60 | 72.64 | 45.05 | 61.40 |
| SmoothQuant | 5.73 | 69.14 | 77.37 | 42.40 | 72.98 | 45.73 | 61.52 |
| GPTQ | 5.41 | 70.56 | 78.56 | 44.40 | 75.04 | 45.98 | 62.91 |
| OmniQuant | 5.43 | 68.43 | 78.29 | 42.00 | 74.58 | 46.16 | 61.89 |
| QuaRot | 7.03 | 65.43 | 76.87 | 40.20 | 71.08 | 41.89 | 59.09 |
| QuaRot$^+$ | 5.57 | 67.79 | 78.18 | 40.60 | 74.62 | 46.67 | 61.57 |
| SpinQuant | 5.20 | 68.98 | 78.45 | 42.80 | 75.63 | 48.04 | 62.78 |
| BRQ | 5.19 | 70.48 | 79.27 | 43.20 | 75.76 | 47.56 | 63.25 |

*Table 9.* Evaluation of different methods on LLaMA-3.2 1B across multiple benchmarks.

| Method | Wiki | WG | PIQA | OBQA | ARC-E | ARC-C | Avg. |
|--------|------|------|------|------|-------|-------|------|
| FP16 | 9.75 | 60.61 | 74.53 | 37.20 | 60.47 | 36.26 | 53.81 |
| BINT4 | 13.56 | 54.78 | 69.26 | 34.80 | 52.74 | 30.20 | 48.36 |
| RTN | 15.91 | 54.30 | 66.10 | 32.80 | 50.37 | 30.88 | 46.89 |
| SmoothQuant | 16.86 | 55.72 | 66.27 | 30.60 | 50.55 | 29.27 | 46.48 |
| GPTQ | 13.35 | 56.66 | 69.58 | 32.40 | 52.48 | 31.48 | 48.52 |
| OmniQuant | 14.32 | 55.09 | 68.12 | 32.80 | 53.37 | 31.48 | 48.17 |
| QuaRot | 17.86 | 55.40 | 66.05 | 29.40 | 47.93 | 28.32 | 45.42 |
| QuaRot$^+$ | 12.78 | 56.74 | 69.85 | 32.60 | 53.42 | 31.56 | 48.83 |
| SpinQuant | 12.72 | 55.38 | 70.67 | 33.00 | 53.87 | 32.51 | 49.09 |
| BRQ | 11.95 | 55.88 | 70.46 | 34.20 | 55.47 | 33.36 | 49.87 |

*Table 10.* Evaluation of different methods on LLaMA-3.2 3B across multiple benchmarks.

| Method | Wiki | WG | PIQA | OBQA | ARC-E | ARC-C | Avg. |
|--------|------|------|------|------|-------|-------|------|
| FP16 | 7.81 | 69.37 | 77.52 | 43.40 | 71.63 | 46.07 | 61.60 |
| BINT4 | 9.29 | 64.64 | 75.19 | 38.60 | 66.50 | 42.41 | 57.47 |
| RTN | 10.27 | 63.93 | 73.06 | 40.00 | 62.16 | 36.94 | 55.22 |
| SmoothQuant | 10.38 | 62.27 | 73.05 | 38.40 | 62.93 | 38.59 | 55.05 |
| GPTQ | 9.50 | 63.29 | 73.55 | 37.00 | 63.04 | 40.10 | 55.40 |
| OmniQuant | 9.85 | 61.01 | 75.84 | 38.40 | 63.30 | 40.27 | 55.76 |
| QuaRot | 13.36 | 59.43 | 70.23 | 35.60 | 57.40 | 35.32 | 51.60 |
| QuaRot$^+$ | 9.92 | 65.43 | 73.18 | 37.80 | 64.65 | 38.48 | 55.91 |
| SpinQuant | 9.85 | 65.11 | 73.61 | 38.60 | 64.87 | 38.74 | 56.19 |
| BRQ | 9.41 | 62.59 | 75.35 | 38.60 | 67.42 | 40.44 | 56.88 |

## .5.2. MISTRAL RESULTS

The following are the detailed experimental results of the Mistral family series models in Tables 1 and 2.

*Table 11.* Evaluation of different methods on Mistral 7B across multiple benchmarks.

| Method | Wiki | WG | PIQA | OBQA | ARC-E | ARC-C | Avg. |
|--------|------|------|------|------|-------|-------|------|
| FP16 | 5.25 | 73.95 | 82.10 | 44.00 | 79.50 | 54.09 | 66.73 |
| BINT4 | 5.63 | 72.06 | 80.41 | 44.80 | 77.57 | 51.54 | 65.28 |
| RTN | 6.56 | 69.06 | 80.73 | 43.00 | 74.66 | 46.84 | 62.86 |
| SmoothQuant | 6.49 | 70.64 | 79.71 | 42.00 | 75.08 | 47.35 | 62.96 |
| GPTQ | 6.00 | 69.53 | 79.48 | 43.20 | 75.71 | 48.80 | 63.34 |
| OmniQuant | 6.37 | 67.25 | 78.56 | 40.20 | 75.13 | 47.95 | 61.82 |
| QuaRot | 6.65 | 68.27 | 78.94 | 39.00 | 71.42 | 44.02 | 60.33 |
| QuaRot$^+$ | 5.73 | 69.61 | 80.84 | 42.80 | 76.85 | 48.20 | 63.66 |
| SpinQuant | 5.68 | 71.90 | 79.71 | 42.80 | 76.35 | 48.21 | 63.79 |
| BRQ | 5.59 | 71.72 | 80.69 | 42.80 | 76.68 | 49.06 | 64.19 |

## .5.3. QWEN RESULTS

The following are the detailed experimental results of the Qwen2.5 family series models in Tables 2.

*Table 12.* Evaluation of different methods on Qwen2.5 1.5B across multiple benchmarks.

| Method | Wiki | WG | PIQA | OBQA | ARC-E | ARC-C | Avg. |
|--------|------|------|------|------|-------|-------|------|
| FP16 | 9.87 | 62.98 | 75.41 | 41.20 | 71.13 | 43.43 | 58.83 |
| BINT4 | 13.98 | 58.43 | 70.44 | 37.00 | 66.67 | 37.36 | 53.98 |
| RTN | 16.61 | 57.93 | 70.51 | 36.60 | 61.20 | 37.20 | 52.69 |
| GPTQ | 13.94 | 58.14 | 70.09 | 37.40 | 62.90 | 37.14 | 53.13 |
| QuaRot | 16.33 | 56.75 | 70.29 | 32.00 | 61.03 | 36.75 | 51.36 |
| QuaRot$^+$ | 12.80 | 58.96 | 71.87 | 36.80 | 62.42 | 37.46 | 53.50 |
| SpinQuant | 12.64 | 59.91 | 71.22 | 36.60 | 62.58 | 37.52 | 53.57 |
| BRQ | 12.15 | 58.96 | 71.87 | 36.80 | 67.00 | 39.51 | 54.83 |

*Table 13.* Evaluation of different methods on Qwen2.5 3B across multiple benchmarks.

| Method | Wiki | WG | PIQA | OBQA | ARC-E | ARC-C | Avg. |
|--------|------|------|------|------|-------|-------|------|
| FP16 | 8.03 | 68.90 | 78.67 | 41.80 | 73.27 | 46.93 | 61.91 |
| BINT4 | 10.32 | 64.25 | 74.61 | 41.20 | 67.17 | 42.92 | 58.03 |
| RTN | 11.03 | 63.46 | 74.10 | 41.00 | 68.48 | 41.81 | 57.77 |
| GPTQ | 10.20 | 64.01 | 75.14 | 39.20 | 70.37 | 42.41 | 58.23 |
| QuaRot | 11.32 | 59.43 | 62.67 | 39.60 | 62.67 | 37.97 | 52.47 |
| QuaRot$^+$ | 9.65 | 63.54 | 75.46 | 38.40 | 70.24 | 43.69 | 58.27 |
| SpinQuant | 9.58 | 63.77 | 75.14 | 40.00 | 72.43 | 44.11 | 59.09 |
| BRQ | 9.48 | 63.38 | 75.52 | 41.80 | 71.97 | 45.73 | 59.68 |

*Table 14.* Evaluation of different methods on Qwen2.5 7B across multiple benchmarks.

| Method | Wiki | WG | PIQA | OBQA | ARC-E | ARC-C | Avg. |
|---|---|---|---|---|---|---|---|
| FP16 | 7.81 | 71.35 | 79.43 | 45.40 | 75.72 | 49.74 | 64.33 |
| BINT4 | 9.07 | 67.01 | 77.15 | 43.80 | 73.06 | 46.50 | 61.50 |
| RTN | 10.00 | 67.80 | 76.01 | 43.60 | 74.79 | 46.93 | 61.83 |
| GPTQ | 9.10 | 65.98 | 77.86 | 44.00 | 75.38 | 47.70 | 62.18 |
| QuaRot | 9.57 | 64.17 | 77.86 | 42.00 | 74.66 | 48.55 | 61.45 |
| QuaRot$^+$ | 8.45 | 65.98 | 76.71 | 42.60 | 74.37 | 49.15 | 61.76 |
| SpinQuant | 8.41 | 66.69 | 78.13 | 43.60 | 76.14 | 49.74 | 62.86 |
| BRQ | 8.34 | 67.96 | 78.40 | 44.80 | 77.02 | 48.89 | 63.41 |

## .5.4. MoE Results

The following are the experimental results of the Mixtral 8×7B. This further confirms the applicability of our findings to the MoE architecture model.

*Table 15.* Evaluation of different methods on Mixtral 8×7B across multiple benchmarks.

| Method | Wiki | WG | PIQA | OBQA | ARC-E | ARC-C | Avg. |
|---|---|---|---|---|---|---|---|
| FP16 | 3.84 | 76.24 | 83.57 | 47.60 | 83.67 | 59.30 | 70.08 |
| RTN | 5.67 | 69.61 | 79.82 | 42.00 | 74.66 | 50.85 | 63.39 |
| QuaRot | 7.56 | 64.17 | 76.12 | 37.80 | 67.72 | 41.64 | 57.49 |
| QuaRot$^+$ | 5.40 | 70.80 | 80.20 | 45.80 | 76.56 | 52.56 | 65.18 |
| BRQ | 4.71 | 71.74 | 80.74 | 46.20 | 78.79 | 53.16 | 66.13 |

## .5.5. Phi Results

The following are the experimental results of the Phi 3 mini. This further confirms the applicability of our findings on the Phi family models.

*Table 16.* Evaluation of different methods on Phi 3 mini across multiple benchmarks.

| Method | Wiki | WG | PIQA | OBQA | ARC-E | ARC-C | Avg. |
|---|---|---|---|---|---|---|---|
| FP16 | 6.02 | 73.72 | 80.63 | 47.80 | 78.70 | 56.57 | 67.28 |
| RTN | 8.62 | 63.38 | 76.28 | 41.40 | 69.74 | 48.72 | 59.90 |
| QuaRot | 46.28 | 49.64 | 57.62 | 28.80 | 42.59 | 28.07 | 41.34 |
| QuaRot$^+$ | 7.96 | 67.25 | 74.54 | 42.80 | 71.46 | 48.46 | 60.90 |
| BRQ | 7.30 | 67.96 | 77.97 | 44.80 | 74.71 | 51.88 | 63.46 |

## .6. The Impact of Quantization/Rotation Block Size on BRQ Performance

Based on LLaMA-3.2 1B, we added a comparative study of BRQ under different quantization/rotation block sizes (16/32/64). The definitions and fusion scheme for R1, R2, and R4 follow SpinQuant.

*Table 17.* BRQ evaluation results across different Quant groups and rotation groups. "QG" represents the quantization group size, and "Rot G" represents the rotation group size.

| Q G | Method | Rot G | Wiki | WG | PIQA | OBQA | ARC-E | ARC-C | Avg. |
|---|---|---|---|---|---|---|---|---|---|
| - | FP16 | - | 9.75 | 60.61 | 74.53 | 37.2 | 60.47 | 36.26 | 53.81 |
| 16 | QuaRot+ | -1 | 12.71 | 56.85 | 69.23 | 31.6 | 54.25 | 32.25 | 48.84 |
| | BRQ | 16 | 12.28 | 57.88 | 70.80 | 31.8 | 53.83 | 32.83 | 49.43 |
| | | 32 | 17.15 | 55.49 | 68.01 | 30.8 | 49.58 | 30.29 | 46.83 |
| | | 64 | 19.46 | 55.41 | 64.47 | 28.6 | 44.36 | 28.60 | 44.29 |
| 32 | QuaRot+ | -1 | 12.78 | 56.74 | 69.85 | 32.6 | 53.42 | 31.56 | 48.83 |
| | BRQ | 16 | 17.33 | 53.67 | 66.92 | 28.2 | 49.37 | 27.82 | 45.20 |
| | | 32 | 11.95 | 55.88 | 70.46 | 34.2 | 55.47 | 33.36 | 49.87 |
| | | 64 | 16.19 | 55.64 | 68.77 | 31.2 | 52.27 | 29.78 | 47.53 |
| 64 | QuaRot+ | -1 | 12.94 | 54.78 | 69.59 | 32.0 | 54.55 | 32.17 | 48.62 |
| | BRQ | 16 | 18.02 | 54.30 | 65.51 | 31.4 | 48.70 | 29.78 | 45.94 |
| | | 32 | 15.35 | 56.20 | 68.17 | 31.6 | 51.52 | 30.20 | 47.54 |
| | | 64 | 11.96 | 56.59 | 70.35 | 33.0 | 56.14 | 33.53 | 49.92 |

The results show that, under these three settings, quantization performance is optimal only when the rotated block size equals the quantization block size. Furthermore, it is significantly better than QuaRot+ using global rotation.

Additionally, it was found that when the block size is too small (block size 16), performance slightly decreases because fewer channels are available for distributing the outlier.

## .7. Ablation Experiments Using a Fusion Rotation Matrix Strategy

We further analyzed the impact of the rotation matrix fusion position on quantization performance, and the experimental results are as follows (where the definitions of R1, R2, and R4 are the same as those of SpinQuant).

*Table 18.* Rotation-position ablation on LLaMA-3.2 1B.

| Rot | Wiki | WG | PIQA | OBQA | ARC-E | ARC-C | Avg. |
|---|---|---|---|---|---|---|---|
| FP16 | 9.75 | 60.61 | 74.53 | 37.20 | 60.47 | 36.26 | 53.81 |
| – | 13.35 | 56.66 | 69.58 | 32.40 | 52.48 | 31.48 | 48.52 |
| R1 | 13.19 | 56.12 | 68.72 | 34.4 | 53.62 | 31.48 | 48.87 |
| R2 | 13.25 | 55.64 | 69.53 | 34.6 | 53.20 | 30.80 | 48.75 |
| R4 | 12.49 | 56.59 | 70.40 | 32.80 | 53.66 | 31.14 | 48.92 |
| R1, R2, R4 | 11.95 | 55.88 | 70.46 | 34.20 | 55.47 | 33.36 | 49.87 |

As can be seen from Table 18, enabling all three (R1+R2+R4) typically yields the best results; even enabling only a subset consistently improves quantization accuracy.

## .8. Comparison of Performance

The prefilling speed test results of BRQ are as Table 19.

*Table 19.* Prefill latency (ms) for LLaMA-2 7B with different sequence lengths and batch sizes. Overhead is calculated relative to MXFP4.

| Batch | Method | SeqLen 512 | | SeqLen 1024 | | SeqLen 2048 | | SeqLen 4096 | |
|---|---|---|---|---|---|---|---|---|---|
| | | Latency | Overhead | Latency | Overhead | Latency | Overhead | Latency | Overhead |
| 1 | MXFP4 | 243.09 | - | 339.31 | - | 584.31 | - | 1274.84 | - |
| | QuaRot | 260.18 | 7.03% | 362.91 | 6.96% | 618.43 | 5.84% | 1334.03 | 4.64% |
| | BRQ | 253.35 | 4.22% | 353.30 | 4.12% | 604.25 | 3.41% | 1307.10 | 2.53% |
| 8 | MXFP4 | 732.34 | - | 1444.37 | - | 3449.58 | - | 8699.71 | - |
| | QuaRot | 789.13 | 7.75% | 1547.36 | 7.13% | 3646.83 | 5.72% | 9032.39 | 3.82% |
| | BRQ | 763.72 | 4.28% | 1499.30 | 3.80% | 3554.16 | 3.03% | 8816.30 | 1.34% |

In addition to the prefilling speed test, we also compared the BRQ generation speed. The specific results are shown in

Table 20.

*Table 20.* Generation speed comparison across LLaMA-2 7B model.

| Batch | Method | SeqLen 128 | | SeqLen 512 | |
|---|---|---|---|---|---|
| | | Latency | Overhead | Latency | Overhead |
| | MXFP4 | 23057.14 | - | 95878.76 | - |
| 8 | QuaRot | 24523.59 | 6.36% | 101809.35 | 6.19% |
| | BRQ | 24053.05 | 4.32% | 99906.44 | 4.20% |

As shown in Table 20 and Table 19, BRQ can effectively reduce inference latency compared to QuaRot, achieving an inference speed closer to that of native MXFP4, regardless of whether it is in the generation or pre-filling stage.

## .9. Relationship Between Rotation Degradation and Quantization Granularity

*Table 21.* Relationship between global rotation degradation and quantization granularity. The experiment used the LLaMA-3.2 1B model.

| Q Group | Method | Wiki | WG | PIQA | OBQA | ARC-E | ARC-C | Avg. |
|---|---|---|---|---|---|---|---|---|
| - | FP16 | 9.75 | 60.61 | 74.53 | 37.2 | 60.47 | 36.26 | 53.81 |
| -1 | RTN | 2408.57 | 51.78 | 51.25 | 27.8 | 27.44 | 23.72 | 36.40 |
| | RTN+Rot | 50.81 | 52.88 | 56.26 | 27.4 | 36.45 | 24.66 | 39.53 |
| 1024 | RTN | 221.22 | 50.20 | 54.24 | 26.4 | 32.58 | 22.70 | 37.22 |
| | RTN+Rot | 40.95 | 50.59 | 59.03 | 25.4 | 38.64 | 26.45 | 40.02 |
| 512 | RTN | 92.82 | 51.78 | 56.42 | 28.6 | 36.20 | 25.51 | 39.70 |
| | RTN+Rot | 34.30 | 52.17 | 58.81 | 25.8 | 39.39 | 25.60 | 40.35 |
| 128 | RTN | 21.56 | 52.96 | 65.56 | 30.6 | 47.77 | 27.47 | 44.87 |
| | RTN+Rot | 20.66 | 54.78 | 65.18 | 29.8 | 47.28 | 28.24 | 45.06 |
| 64 | RTN | 15.76 | 53.83 | 66.43 | 31.6 | 50.00 | 30.80 | 46.53 |
| | RTN+Rot | 17.97 | 53.28 | 68.34 | 31.4 | 48.48 | 29.35 | 46.17 |
| 32 | RTN | 13.56 | 54.78 | 69.26 | 34.8 | 52.74 | 30.20 | 48.36 |
| | RTN+Rot | 15.28 | 55.49 | 67.25 | 30.6 | 50.34 | 29.52 | 46.64 |

Our comparison of global rotation and RTN with different quantization group sizes of BINT4 using FP16 scale reveals a more detailed picture (as shown in Table 21). When the quantization group is large (per-channel, 1024, 512, 128), global rotation consistently improves quantization accuracy. In contrast, when the group size becomes small (64, 32), global rotation actually degrades performance. This is because when the quantization granularity is coarse and the group size is large, the proportion of regular blocks is small, and their quantization error growth is masked by the loss reduction of outlier blocks. This is also the main reason why using grouped quantization in previous INT4 quantizations did not lead to model accuracy degradation.

## .10. BRQ with NVFP4

We have additionally evaluated BRQ under the NVFP4 format, which is supported by the latest NVIDIA GPUs and employs E4M3 scaling instead of E8M0. The corresponding results are provided in Table 22.

*Table 22.* Performance comparison on LLaMA-3.2 models under different NVFP4 quantization methods. BRQ consistently maintains or improves average accuracy.

| Model | Method | Wiki | WG | PIQA | OBQA | ARC-E | ARC-C | Avg. |
|---|---|---|---|---|---|---|---|---|
| | FP16 | 9.75 | 60.61 | 74.53 | 37.20 | 60.47 | 36.26 | 53.81 |
| | NVFP4 + RTN | 11.74 | 56.91 | 72.03 | 35.60 | 56.86 | 32.85 | 50.85 |
| llama 3.2 1B | NVFP4 + GPTQ | 11.45 | 57.62 | 71.55 | 35.20 | 56.56 | 33.62 | 50.91 |
| | NVFP4 + QuaRot$^+$ | 12.26 | 58.64 | 73.29 | 32.00 | 54.84 | 32.85 | 50.32 |
| | NVFP4 + BRQ | 11.30 | 59.51 | 71.22 | 35.20 | 56.94 | 32.68 | 51.11 |
| | FP16 | 7.81 | 69.37 | 77.52 | 43.40 | 71.63 | 46.07 | 61.60 |
| | NVFP4 + RTN | 8.63 | 65.27 | 76.22 | 41.20 | 69.95 | 43.86 | 59.30 |
| llama 3.2 3B | NVFP4 + GPTQ | 8.56 | 65.59 | 76.44 | 41.40 | 70.66 | 43.86 | 59.59 |
| | NVFP4 + QuaRot$^+$ | 8.54 | 66.66 | 76.33 | 41.60 | 69.65 | 43.79 | 59.61 |
| | NVFP4 + BRQ | 8.49 | 67.01 | 76.44 | 40.60 | 70.72 | 43.58 | 59.67 |

As shown in Table 22, although NVFP4 + RTN already yields relatively small quantization error, BRQ still brings consistent and clear improvements when applied on top of NVFP4. This demonstrates that our method remains effective and does not rely on any specific scaling format, and thus is robust to future hardware evolution.

## .11. BRQ with Other Formats

To verify the applicability of BRQ to other data formats, we extended it to BINT4, BFP4, MXINT4, MXINT6, MXFP6 and MXINT8. The experiment used LLaMA-3.2 1B, with a group size of 32 for all bit widths, and PoT scale was used for the MX format.

As shown in Table 23, the RTN precision of MXINT8 is almost identical to that of FP16, so further optimization is not very meaningful. In experiments with lower bit widths ($< 8$ bits), the results are consistent with those of MXFP4; global rotations degrade quantization performance, and BRQ consistently exhibits the best quantization performance. This confirms that our findings are applicable to different data types.

*Table 23.* Evaluation of different quantization formats. The experiment used the LLaMA-3.2 1B model.

| Format | Method | Wiki | WG | PIQA | OBQA | ARC-E | ARC-C | Avg. |
|---|---|---|---|---|---|---|---|---|
| FP16 | - | 9.75 | 60.61 | 74.53 | 37.2 | 60.47 | 36.26 | 53.81 |
| | RTN | 9.76 | 60.38 | 74.48 | 37.4 | 60.31 | 36.26 | 53.77 |
| MXINT8 | QuaRot | 9.78 | 60.69 | 74.37 | 37.0 | 60.35 | 36.01 | 53.68 |
| | QuaRot+ | 9.77 | 60.06 | 74.27 | 37.0 | 60.65 | 36.35 | 53.67 |
| | BRQ | 9.76 | 61.17 | 74.54 | 37.0 | 60.40 | 36.35 | 53.89 |
| | RTN | 10.08 | 60.01 | 74.10 | 36.0 | 59.78 | 36.26 | 53.23 |
| MXINT6 | QuaRot | 10.10 | 59.83 | 74.05 | 35.6 | 59.68 | 36.35 | 53.10 |
| | QuaRot+ | 9.92 | 59.59 | 73.78 | 36.6 | 60.14 | 36.28 | 53.28 |
| | BRQ | 9.89 | 60.14 | 74.21 | 37.8 | 60.27 | 36.52 | 53.79 |
| | RTN | 19.56 | 53.43 | 65.02 | 29.6 | 44.82 | 28.67 | 44.31 |
| MXINT4 | QuaRot | 26.19 | 54.46 | 61.48 | 28.6 | 45.92 | 29.86 | 44.06 |
| | QuaRot+ | 13.99 | 54.22 | 67.46 | 32.8 | 51.22 | 30.63 | 47.27 |
| | BRQ | 12.40 | 56.75 | 69.75 | 32.0 | 53.83 | 31.40 | 48.75 |
| | RTN | 9.93 | 60.38 | 74.76 | 36.6 | 60.02 | 35.58 | 53.47 |
| MXFP6 | QuaRot | 10.09 | 60.06 | 73.94 | 35.8 | 59.43 | 36.52 | 53.15 |
| | QuaRot+ | 9.91 | 60.54 | 74.92 | 36.4 | 60.23 | 35.75 | 53.57 |
| | BRQ | 9.89 | 61.17 | 74.32 | 37.8 | 60.73 | 37.29 | 54.26 |
| | RTN | 13.56 | 54.78 | 69.26 | 34.8 | 52.74 | 30.20 | 48.36 |
| BINT4 | QuaRot | 15.27 | 56.67 | 65.89 | 31.4 | 49.24 | 30.89 | 46.82 |
| | QuaRot+ | 11.66 | 55.64 | 72.52 | 34.0 | 56.99 | 33.36 | 50.50 |
| | BRQ | 11.48 | 56.20 | 72.00 | 36.6 | 57.20 | 33.25 | 51.05 |
| | RTN | 12.35 | 57.22 | 70.95 | 34.6 | 54.80 | 31.74 | 49.86 |
| BFP4 | QuaRot | 17.04 | 54.06 | 66.54 | 31.4 | 49.07 | 31.40 | 46.49 |
| | QuaRot+ | 12.05 | 56.83 | 71.16 | 34.2 | 55.93 | 32.94 | 50.21 |
| | BRQ | 11.11 | 56.99 | 71.11 | 35.4 | 56.73 | 34.39 | 50.92 |

### .12. Relation to Concurrent and Recent Block-level Methods

Recent and concurrent studies have also explored local or block-level transformations for efficient low-bit quantization. These works are closely related to ours in that they all suggest that transformation granularity is important under group-wise or microscaling quantization formats. However, their motivations and emphases are different from ours.

LightRot (Kim et al., 2025) adopts local/grouped rotations mainly to improve the robustness and hardware efficiency of online rotation transforms. Its design is motivated by the dimension sensitivity and implementation cost of fast Hadamard transforms under practical inference settings. In contrast, our work focuses on the MXFP4-specific failure mechanism of global rotation: global rotation redistributes outlier energy across quantization blocks, inflates the scales of many regular blocks, and thus interacts poorly with MXFP4's block-wise scaling. Therefore, BRQ is derived as a granularity-aligned correction to this failure mode, rather than as a general acceleration strategy for online transforms.

MR-GPTQ (Egiazarian et al., 2025) is another concurrent work that studies FP4/MXFP4 quantization and includes block-wise Hadamard transforms together with additional system-oriented components. Its main emphasis is an FP4-specialized quantization and inference stack. Our work is complementary: we isolate and analyze why global rotation becomes harmful under MXFP4, and show that block-aligned rotation is the key minimal correction for restoring the compatibility between rotation-based PTQ and MXFP4. The empirical observations in MR-GPTQ are consistent with our analysis, as block-wise transformations play an important role under MXFP4-style quantization.

Very recent works such as MixQuant (Sanjeet et al., 2026) and BATQuant (Li et al., 2026) further explore stronger block-level transformation or mixed-format designs. These methods can be viewed as complementary extensions of the same general principle: the transformation granularity should be aligned with the quantization/scaling granularity. Our goal is not to exhaustively compare against all such concurrent or follow-up systems, which often differ in optimization objectives, kernels, and deployment assumptions. Instead, our contribution is to identify the root cause of the incompatibility between global rotation and MXFP4, and to validate block-aligned rotation as a simple and effective correction derived from this mechanism.

Overall, these related studies reinforce the importance of granularity-aware transformation design for emerging low-precision formats. Compared with them, our paper emphasizes the failure-mode analysis and the minimality of the correction: once the cross-block scale inflation caused by global rotation is identified, restricting rotation within quantization blocks becomes a natural and practical remedy.

