# OpenReview forum: "Block Rotation is All You Need for MXFP4 Quantization"
_ICML.cc/2026/Conference — ICML 2026 regular_

### Official Review · Reviewer_Tat8 · 2026-03-12

**Soundness:** 2
**Presentation:** 3
**Significance:** 3
**Originality:** 2
**Overall Recommendation:** 4
**Confidence:** 5

**Summary:**

This paper studies post-training W4A4 quantization under the MXFP4 format. It shows that global rotation, which is often helpful in INT4, interacts badly with MXFP4 because PoT block scaling makes many regular blocks harder to quantize after rotation. The proposed BRQ method replaces global rotation with block-wise rotation aligned to quantization groups, and it improves over QuaRot+/SpinQuant on several LLaMA, Mistral, and Qwen models while also reducing rotation overhead.

**Compliance With Llm Reviewing Policy:**

Affirmed.

**Final Justification:**

I maintain my score at present.

**Key Questions For Authors:**

1. Can you provide a more controlled comparison where competing methods use matched calibration and optimization budgets?
2. How much of BRQ’s gain comes from block alignment itself, versus simply reducing the rotation dimension?
3. Please reconcile the claim in Section 5.1 with the Mistral-7B result in Table 2.

**Limitations:**

Yes.

**Strengths And Weaknesses:**

The paper addresses a timely and practical problem, since MXFP4 is increasingly relevant for hardware-efficient LLM deployment. The empirical study is fairly broad, covering multiple PTQ families, several model families, and useful ablations on rotation size, rotation position, other formats, and latency. The main intuition is clear, and the block-scale analysis gives a plausible explanation for why global rotation fails under MXFP4.

The main weakness is that the novelty is moderate. Once the mismatch between global rotation and block-wise PoT scaling is identified, block-aligned rotation is a fairly natural fix. In addition, the benchmark is not perfectly controlled, and it seems different methods use different calibration or optimization budgets. I also noticed an overstatement in Section 5.1: the text says BRQ surpasses BINT4 in all cases except LLaMA-3.2 3B, but Table 2 shows BRQ is also below BINT4 on Mistral-7B average accuracy.

---

> ### Author Rebuttal · Authors · 2026-03-28
>
> Thank you for the helpful comments.
>
> > W1: Novelty.
>
> While BRQ is a concise structural change in form, the core contribution of this paper is not merely “changing global rotation to block-wise rotation.” Rather, by systematically comparing PTQ methods under a unified MXFP4 setting, we reveal and explain a previously underexplored failure mode: global rotation increases the scales of many regular blocks, while MXFP4’s PoT block scaling further amplifies the resulting quantization error. BRQ is thus a direct correction motivated by this mechanism, rather than an ad hoc tweak. In other words, this paper not only identifies the limitation of global rotation under MXFP4, but also proposes a simple and effective remedy aligned with quantization granularity. More importantly, we believe that the contribution of this paper is not limited to the specific method of BRQ, but also provides clear inspiration for further improvements to block-wise/local rotation: the design of rotation should match the quantization/scaling granularity. At the same time, we also want to emphasize that for PTQ problems geared towards practical deployment, a simple and effective solution is itself an important advantage.
>
> > W2/Q1: Matched budget.
>
> We agree that a controlled matched-budget comparison is valuable. We therefore added a comparison on LLaMA-2 7B where all methods use the same budget: 128 sequences of length 2048. The main conclusion remains unchanged: global rotation remains unstable under MXFP4, while BRQ/BRQ-Spin still clearly outperform the corresponding global-rotation baselines.
>
> |Method|PPL|WG|PIQA|OBQA|ARC-E|ARC-C|Avg|
> |-|-|-|-|-|-|-|-|
> |FP16|5.47|68.98|79.05|44.20|74.57|46.16|62.59|
> |SmoothQuant|7.04|64.64|76.17|39.00|66.75|39.33|57.18|
> |GPTQ|6.56|66.61|76.55|40.60|71.12|41.46|59.27|
> |OmniQuant|6.56|63.06|76.33|36.60|67.09|40.27|56.67|
> |QuaRot+|6.29|67.24|75.68|39.80|69.02|40.01|58.35|
> |BRQ|5.84|67.09|76.77|44.80|73.23|43.17|61.01|
> |SpinQuant|5.83|67.40|77.26|40.00|68.48|39.93|58.61|
> |BRQ_{Spin}|5.82|68.90|77.69|43.40|72.77|43.77|61.31|
>
> This controlled comparison supports that our main conclusion is not driven by unequal calibration/optimization budgets.
>
> > Q2: Alignment vs. dimension.
>
> Existing experiments support a more precise conclusion: reducing the rotation dimension helps because it limits cross-block outlier spreading, but smaller is not always better. Sec. 5.2 shows that large rotation groups amplify outlier impact, while overly small groups provide insufficient channels for redistribution. Consistently, Figure 8 and Appendix Table 16 show that the best results occur when the rotation block size matches the quantization block size; both smaller and larger groups degrade. Thus, the benefit comes from a better trade-off, and matching is the most robust default choice in our setting.
>
> > W3/Q3: Claim correction.
>
> The reviewer is correct: on Mistral-7B, BRQ average accuracy is below BINT4, so the text saying “except LLaMA-3.2 3B” is inaccurate. We will correct this to a more precise statement, e.g., BRQ delivers substantial improvements, surpassing the strong BINT4 baseline on most models. This wording issue does not affect the main conclusion.

---

> > ### Author Rebuttal · Reviewer_Tat8 · 2026-04-01
> >
> > With the exception of clarifying the novelty of this work, the authors have satisfactorily addressed all other concerns I raised.

---

> > > ### Author Response · Authors · 2026-04-02
> > >
> > > Thank you for the follow-up.
> > >
> > > We believe the main contribution of this paper is to reveal and explain a previously underexplored failure mode of rotation-based PTQ under MXFP4. Based on this analysis, we propose BRQ as a direct correction aligned with the quantization granularity. Beyond the specific method itself, we hope this mechanism-level understanding can inspire further, more novel method-level improvements and help advance PTQ research for MXFP4 and related grouped formats.

---

### Official Review · Reviewer_rhzu · 2026-03-12

**Soundness:** 3
**Presentation:** 3
**Significance:** 2
**Originality:** 3
**Overall Recommendation:** 5
**Confidence:** 4

**Summary:**

This paper studies the incompatibility between block-wise power-of-two (PoT) scaling in MXFP4 and conventional global rotation-based PTQ for LLMs. The authors show that, under MXFP4, global rotation can spread outlier energy from a few channels to many regular blocks, which increases quantization error instead of reducing it. Based on this observation, they propose Block-wise Rotation Quantization (BRQ), which performs rotation only within the same granularity as the quantization blocks. The paper further presents a benchmark-style comparison of representative PTQ methods under MXFP4 and shows that BRQ improves perplexity and zero-shot accuracy over prior rotation-based baselines, while also reducing online rotation overhead.

**Compliance With Llm Reviewing Policy:**

Affirmed.

**Final Justification:**

The authors have addressed my main concerns, and I will raise my score from 4 to 5.

**Key Questions For Authors:**

1. The paper argues that BRQ is more compatible with MXFP4 than global rotation, but how does it compare more directly with other local or block-level transformation strategies beyond the specific baselines considered here?
2. The claimed benchmark contribution seems somewhat broader than what is actually provided. Could the authors clarify whether this is intended as a reusable benchmark artifact or mainly as a unified experimental comparison under MXFP4?
3. Since one practical motivation is reduced runtime overhead, it would be helpful to provide a more complete discussion of end-to-end efficiency, especially relative to other system-level approaches.

**Limitations:**

This paper provides a useful analysis of why global rotation conflicts with MXFP4 quantization, but the proposed solution is still fairly format-specific. The effectiveness of BRQ depends strongly on the block-wise PoT scaling structure of MXFP4, so its generality beyond this setting remains unclear.

A second limitation is that the core method is conceptually simple: it mainly reduces the rotation granularity from global to block-wise. While this is practically useful, it also means the methodological novelty is somewhat limited, especially when compared to related work that explores similar block-level rotation ideas in MXFP4/NVFP4 settings.

Finally, although the paper reports some runtime benefit, the efficiency analysis remains narrower than a full system-level evaluation. The work would be stronger with a clearer positioning against related approaches that also include optimized kernels and more complete end-to-end runtime analysis.

**Strengths And Weaknesses:**

### 1. Soundness

Strengths

The paper provides a clear empirical observation that rotation behaves very differently under MXFP4 than under standard INT4-style settings. The analysis connecting global rotation, block-wise PoT scaling, and the growth of regular-block quantization error is intuitive and reasonably supported by visualizations, ablations, and downstream results. The proposed BRQ method is also simple and practically motivated, and the experiments across multiple LLMs and MoE models make the empirical trend fairly convincing.

Weaknesses

The main weakness is that the proposed method is relatively close to a structural modification of existing rotation-based quantization: it mainly restricts the rotation granularity from global to block-wise. As a result, the methodological novelty is somewhat limited. In addition, while the paper argues that BRQ is both more accurate and more efficient, the runtime evidence is still relatively limited compared with broader system-level evaluations.

### 2. Presentation

Strengths

The paper is generally well organized. The benchmark, failure-mode analysis, and proposed fix are presented in a natural sequence, and the core message is easy to follow. The visual illustrations of how global rotation and block-wise rotation affect outlier distribution are particularly helpful.

Weaknesses

Some contribution claims feel slightly overstated. In particular, the claimed “benchmark” is closer to a unified comparison of representative methods under a specific setting than to a new benchmark artifact or infrastructure. Also, the paper would benefit from a clearer and more explicit discussion of how it differs from closely related and follow-up works using block-level rotation ideas.

### 3. Significance

Strengths

This study's main area concerns low-bit PTQ for emerging MXFP4 hardware formats, which is a practically important topic given growing hardware support for MX formats. The paper identifies a useful and actionable lesson: quantization design should align with the scaling granularity of the underlying hardware format. This is a relevant insight for both researchers and practitioners working on low-bit LLM deployment.

Weaknesses

At the same time, the significance is somewhat constrained by the fact that the method is strongly tied to MXFP4’s block-wise PoT scaling structure. Its broader usefulness outside MXFP4-style settings is not yet clear. The paper intends to consider a notable area, but the practical impact may remain somewhat specialized unless the idea generalizes beyond this format family.

### 4. Originality

Strengths

The originality of the paper lies more in the failure-mode analysis than in the algorithm itself. The paper offers a useful explanation of why global rotation, which is often beneficial in INT4 quantization, can become harmful in MXFP4. This insight is valuable and helps clarify the design space of PTQ under emerging low-precision formats.

Weaknesses

However, the actual method is not highly novel algorithmically. BRQ is essentially a block-wise adaptation of existing rotation-based quantization, rather than a fundamentally new quantization principle. In light of related and follow-up work that also considers block-level rotation in MXFP4/NVFP4 settings, the distinctiveness of the method is not yet fully compelling.

---

> ### Author Rebuttal · Authors · 2026-03-28
>
> Thank you for the constructive comments.
>
> > Soundness/Originality: Core contribution.
>
> We agree that BRQ is a concise structural change in form; however, we want to emphasize that the core contribution of this paper lies first in discovering and systematically revealing a previously underrecognized failure mode in MXFP4: global rotation enlarges the scales of many regular blocks, and MXFP4’s PoT scaling further amplifies the resulting quantization error, and further in providing an effective correction scheme aligned with quantization granularity. In other words, the importance of this paper lies not only in proposing BRQ, but also in clarifying why global rotation becomes unreliable under MXFP4 and how a granularity-aligned correction can address this failure mode. At the same time, we also want to emphasize that for PTQ problems geared towards practical deployment, a simple and effective solution is itself a significant advantage. BRQ does not rely on complex optimizations, yet it can stably improve the compatibility of rotation-based quantization under MXFP4, which is its practical value.
>
> > Presentation/Q2: Benchmark wording.
>
> We thank the reviewer for pointing this out and agree that the term “benchmark” may be misleading here. Our intent was not to claim a benchmark artifact or infrastructure, but rather a unified/systematic comparison of representative PTQ methods under MXFP4. We will revise the wording in the abstract, introduction, and contributions accordingly.
>
> > Presentation/Q1: Related work.
>
> We thank the reviewer for this suggestion and agree that the related-work discussion should be refined. In particular, we will more explicitly position our work relative to recent concurrent and closely related studies on block-/local-granularity transformations for MXFP4-style quantization.
>
> The distinction from earlier work such as LightRot is that its local/grouped rotation is motivated mainly by improving the robustness and hardware efficiency of online transforms, especially given the dimension sensitivity of FHT and its limitations on some distributions. Our focus here is different: we study the MXFP4-specific failure mechanism of global rotation and derive block-aligned rotation as a correction that matches the quantization granularity.
>
> The concurrent work MR-GPTQ also considers MXFP4/FP4 quantization with block-wise Hadamard transforms and additional system-oriented components. Its emphasis is different from ours: MR-GPTQ mainly focuses on an FP4-specialized engineering/system stack, whereas our paper focuses on explaining why global rotation fails under MXFP4 and on BRQ as a minimal correction aligned with quantization granularity. Its ablation results are nevertheless empirically consistent with our observations in that block-aligned rotation plays an important role under MXFP4-style quantization, while reordering provides only incremental additional gains in their reported setting (MR-GPTQ, Fig. 15).
>
> We will also briefly discuss very recent follow-up works built on similar block-level ideas, such as MixQuant and BATQuant. We view these as complementary extensions rather than contradictory evidence: they further explore how block-aligned transformations can be strengthened, while our paper isolates and validates the core mechanism that makes quantization-granularity-aligned rotation effective in the first place.
>
> Since these concurrent/follow-up methods are not centered on the same question as ours, we view them primarily as related work rather than directly matched baselines. We will revise the related-work section accordingly and clarify that our goal is to explain the incompatibility of global rotation with MXFP4 and show that block-aligned rotation is a simple and effective correction.
>
> > Significance/Limitation: Generality.
>
> Although the main focus is MXFP4, we already evaluated broader formats in the appendix. Appendix A.10 shows stable gains under NVFP4, and Appendix A.11 shows consistent trends under BINT4, BFP4, MXINT4, and MXFP6. This suggests that the underlying trend may extend beyond one specific scale encoding, more broadly to grouped quantization settings where shared block-wise scaling interacts with rotation granularity.
>
> > Q3: Runtime.
>
> We agree that the original efficiency evidence mainly reflects relative online rotation overhead rather than a full end-to-end system evaluation. First, BRQ adds essentially no extra PTQ cost: relative to QuaRot, it only changes the random Hadamard initialization, and BRQ-Spin adds no extra optimization or fusion cost over SpinQuant. Second, BRQ is more efficient than global rotation after PTQ: it reduces online complexity from O(N^2) to O(N×32), and our added end-to-end results on RTX 5090 + LLaMA-3.2 1B show 2%–6% lower prefill latency than QuaRot, with extra overhead over native MXFP4 reduced from 3.7%/7.4% to 1.5%/0.8%.
>
> |Prefill(ms)| 256|512|
> |-|-|-|
> |BF16|100.60|201.77|
> |MXFP4|53.25|89.24|
> |QuaRot|55.20|95.85|
> |BRQ|54.03|89.91|

---

> > ### Author Rebuttal · Reviewer_rhzu · 2026-04-03
> >
> > The authors have addressed my main concerns, and I will raise my score from 4 to 5.

---

> > > ### Author Response · Authors · 2026-04-03
> > >
> > > Thank you for the follow-up and for the careful consideration of our rebuttal. We are glad that our response helped address your concerns, and we sincerely appreciate your time and constructive feedback throughout the review process.

---

### Official Review · Reviewer_HxS1 · 2026-03-13

**Soundness:** 3
**Presentation:** 4
**Significance:** 3
**Originality:** 3
**Overall Recommendation:** 4
**Confidence:** 4

**Summary:**

The paper shows that rotation methods that help INT4 quantization can hurt MXFP4 because they clash with MXFP4's blockwise power-of-two scaling. It proposes BRQ, a blockwise rotation method that matches the quantization blocks, and demonstrates better accuracy and lower overhead than global rotation baselines across several LLMs.

**Compliance With Llm Reviewing Policy:**

Affirmed.

**Final Justification:**

My main concerns are only partially resolved. I still view the theoretical analysis as not rigorous enough. And I remain unconvinced that the method itself is sufficiently novel beyond a natural block-aligned correction, especially in the absence of a theoretical justification for aligning the block size with the quantization group size.

**Key Questions For Authors:**

How would the findings in this paper generalize to the NVFP4 format, which is similar to MXFP4 but has a smaller block size (16) and a different scaling format (E3M4)?

**Limitations:**

The title is somewhat overclaiming. While blockwise rotation clearly improves MXFP4 quantization, the results still leave a noticeable gap to full-precision performance, and the paper does not establish that block rotation is the unique or optimal solution for this setting. I suggest changing the title to better reflect the actual scope of this research.

**Strengths And Weaknesses:**

**Strengths**

1. The paper studies a practical question of whether existing PTQ methods transfer well to MXFP4. This is relevant for efficient LLM deployment.

2. The experiments are fairly broad across baselines, models, and metrics. That gives the main conclusions reasonable empirical support.

3. The paper gives a clear explanation for why global rotation hurts MXFP4. The failure mode is intuitive and well motivated.

4. The proposed fix is simple and effective. Blockwise rotation is easy to understand and improves over global rotation baselines.

---

**Weaknesses**

1. The idea of block rotation seems a bit trivial, which limits the novelty of this paper.

2. The theoretical analysis is helpful but not fully rigorous. It explains the observed behavior, but it does not provide especially tight guarantees for MXFP4.

- Section 4.3 Proof 2 uses a uniform min-max quantization scheme to say that a larger scale leads to larger MSE, but MXFP4 uses a non-uniform FP4 code together with power-of-two scaling. That means the exact error behavior is more complicated than the proof suggests, so the derivation is not really a precise guarantee for MXFP4 itself.

- The analysis in Section 4.3 mainly explains why global rotation can fail, but it does not fully prove when or how much blockwise rotation should help. For example, it does not derive a tight bound comparing global and blockwise rotation under MXFP4, nor does it formally justify why matching the rotation size to the quantization block size should be optimal. That conclusion is supported empirically in Section 5.2, but not established theoretically.

3. The inference runtime results (Tables 18, 19) are useful, but they rely on simulation rather than native FP4 hardware. However, the concurrent work [1], which shows similar findings on the MXFP4/NVFP4 global rotation problems and blockwise rotation solutions, provides dedicated FP4 CUDA kernels.

---

References

[1] Egiazarian et al. Bridging the Gap Between Promise and Performance for Microscaling FP4 Quantization. ICLR 2026. URL https://openreview.net/forum?id=zCBGe9AqJZ

---

> ### Author Rebuttal · Authors · 2026-03-28
>
> Thank you for the constructive comments.
>
> > W1: The idea is trivial.
>
> We agree that BRQ is simple in form. However, the main contribution is not merely replacing global rotation with block-wise rotation, but systematically studying PTQ behavior under MXFP4, identifying that global rotation enlarges the scales of many regular blocks and that MXFP4’s PoT scaling further amplifies the resulting error, and deriving a granularity-aligned correction principle from that analysis. At the same time, we also want to emphasize that for practical deployment problems like PTQ, a simple and effective solution is itself a significant advantage. BRQ does not rely on complex optimizations, yet it can stably improve the compatibility of rotation-based quantization under MXFP4, which is its practical value. More importantly, we believe this analysis can guide future PTQ designs and inspire more deliberate solutions for MXFP4 and related grouped formats.
>
> > W2: Theory scope.
>
> Proof 2 is intended as a trend-level explanation rather than a tight MXFP4-specific bound. This is consistent with the goal of the paper: our purpose is not to derive a complete MXFP4 error theory, but to explain a previously underexplored failure mode — why global rotation systematically breaks under MXFP4. For this purpose, the key point is the causal trend that global rotation enlarges many regular-block scales, and that larger block scales increase quantization error under block-wise scaling. This conclusion is not supported by Proof 2 alone: Figure 3 directly visualizes the growth of PoT rounding/block quantization error with scale, and Figure 6 verifies the significant increase of regular-block error after global rotation. Thus, Proof 2 provides the mechanism-level explanation, while Figures 3 and 6 provide MXFP4-specific validation. For the goal of this paper — explaining the failure mode rather than deriving a complete MXFP4 error theory — we believe this level of theory is appropriate and sufficient.
>
> We acknowledge that the current theory mainly explains the failure mode of global rotation, while the superiority of BRQ — especially the conclusion that rotation block size should align with quantization block size — is currently supported by consistent empirical evidence rather than a formal optimality proof. However, this is fully consistent with our design logic: BRQ is not an arbitrary heuristic, but a direct correction derived from the identified mechanism. If the harm comes from cross-block scale inflation induced by global rotation under block-wise quantization, then aligning rotation granularity with quantization granularity is the most natural fix. In this sense, Section 4 explains why BRQ should help, while Section 5.2 and Appendix Table 16 verify that this design indeed works. For our paper, this “mechanism → design motivation → empirical validation” chain is sufficient to support the main claim, even though we do not present it as a formal optimality theorem.
>
> > W3: Runtime.
>
> To strengthen this point, we added end-to-end results on RTX 5090 + LLaMA-3.2 1B: compared with QuaRot, BRQ further reduces prefill latency by about 2%–6%, and reduces the extra overhead over native MXFP4 from about 3.7%/7.4% to about 1.5%/0.8%. Thus, BRQ is not only better than global rotation, but also remains near-native MXFP4 efficient. We also thank the reviewer for pointing out the concurrent work [1], which also motivated us to strengthen the end-to-end efficiency evidence; we will further discuss this line of work in the revision. Thus, BRQ is not only better than global rotation, but also remains near-native MXFP4 efficient.
>
> |Prefill(ms)| 256|512|
> |-|-|-|
> |BF16|100.60|201.77|
> |MXFP4|53.25|89.24|
> |QuaRot|55.20|95.85|
> |BRQ|54.03|89.91|
>
> > Q: Generalization.
>
> Appendix A.10 shows that BRQ also gives consistent gains under NVFP4 on LLaMA-3.2 1B/3B. Appendix A.11 further shows consistent trends on BINT4, BFP4, MXINT4, MXINT6, and MXFP6. This suggests that the underlying trend may extend beyond one specific scale encoding, more broadly to grouped quantization settings where shared block-wise scaling interacts with rotation granularity.
>
> > Limitation: Title.
>
> Our intent was to emphasize the importance of block-aligned rotation for MXFP4, not to claim that it is the unique or provably optimal solution for this setting. We are happy to revise the title to a more cautious version.

---

> > ### Author Rebuttal · Reviewer_HxS1 · 2026-04-03
> >
> > Thank you for the response! I will maintain my rating.
> >
> > My main concerns are only partially resolved. I still view the theoretical analysis as not rigorous enough. And I remain unconvinced that the method itself is sufficiently novel beyond a natural block-aligned correction, especially in the absence of a theoretical justification for aligning the block size with the quantization group size.

---

> > > ### Author Response · Authors · 2026-04-04
> > >
> > > Thank you for the follow-up and for the continued consideration of our paper. We are glad that our rebuttal helped clarify part of your concerns.
> > >
> > > Regarding the theory, our intent is for the current analysis to serve as a mechanism-level explanation, rather than a formal optimality proof for BRQ or for block-size selection. In the revision, we will make this scope more explicit and better reflect the role of the analysis.
> > >
> > > Regarding novelty, we believe the main contribution of this paper is to reveal and explain a previously underexplored failure mode of rotation-based PTQ under MXFP4. We hope this understanding can inspire further method-level improvements and help advance PTQ research in this area.

---

### Official Review · Reviewer_Ke7E · 2026-03-19

**Soundness:** 3
**Presentation:** 4
**Significance:** 3
**Originality:** 2
**Overall Recommendation:** 5
**Confidence:** 4

**Summary:**

The paper studies various PTQ methods for LLMs under the recently emerging MXFP4 format. The authors analyze a range of existing methods and find that rotation-based approaches, despite their success in integer quantization, perform particularly poorly in MXFP4. They identify the root cause as a mismatch between MXFP4’s block-wise power-of-two scaling and global rotation, which redistributes outlier energy and increases quantization error. To address this issue, they propose a fine-grained block-wise rotation (BRQ) that confines rotation within each quantization block. This approach significantly improves accuracy across multiple models and outperforms prior methods.

**Compliance With Llm Reviewing Policy:**

Affirmed.

**Final Justification:**

My main concerns have been largely addressed. While the proposed method appears relatively straightforward and the level of originality is somewhat limited, I believe the paper’s overall contribution meets the bar.

**Key Questions For Authors:**

- Why do rotation-based quantization methods outperform others in INT formats, while showing inconsistent or poor performance in other settings? For example, in Figure 2, why does applying rotation alone degrade performance even for BINT4, which has precise scaling?
- What are the limitations of block-wise rotation compared to global rotation? In particular, can it effectively handle extreme outlier cases?
- Should the rotation block size always match the quantization block size? Under what conditions might different choices be preferable?

**Limitations:**

What are the expected limitations of block-wise rotation compared to global rotation? In particular, can it effectively handle extreme outlier cases?

**Strengths And Weaknesses:**

### Strengths
- Clear and important motivation.
- Well-organized introduction and overall structure that consistently highlights a key insight: the mismatch between block-wise PoT scaling and global rotation. The idea is intuitive and the solution directly targets it.
- The paper shows strong empirical evidence and an analytical breakdown of MXFP4 behavior. The paper clearly explains how PoT scaling struggles with large magnitudes and how rotation redistributes values, increasing overall scales. Theoretical analysis (e.g., scale vs. error relationship) is insightful and well-aligned with experiments.
- The proposed method is simple, practical, and demonstrates consistent performance improvements across models.

### Weakness
- The analysis and claims rely on the assumption that rotation-based methods are strong baselines under non-PoT or more precise scaling, but this is not thoroughly validated across diverse settings. For example, even in INT4 settings, the effectiveness of rotation depends on the quantization scheme (e.g., per-channel vs group-wise scaling), and can sometimes degrade performance (as partially observed in BINT4 in Figure 2). This raises questions about whether rotation is consistently a strong baseline under non-PoT or more precise scaling schemes.
- Limited discussion on the fundamental trade-off between global and block-wise rotation (e.g., reduced expressivity or limited ability to handle extreme outliers), particularly in cases where outliers span across blocks or require global redistribution.
- Although simple and practical, the solution is incremental in nature, as it mainly adapts existing rotation techniques rather than introducing a fundamentally new paradigm.
- Practical overhead is only partially discussed. While rotation is reduced, there is still additional complexity in integration, tuning, and block alignment, which may affect real deployment.

---

> ### Author Rebuttal · Authors · 2026-03-29
>
> Thank you for the constructive comments.
> > W1/Q1: Rotation baselines.
>
> We view rotation-based methods as strong baselines under **non-block** activation quantization(conventional INT4), as they are widely used directions in several recent SOTA W4A4/ultra-low-bit PTQ methods (SpinQuant,OSTQuant). Our point is not that PoT alone causes global rotation to fail. Rather, for **block quantization**, global rotation first inflates the scales of the many regular blocks, which increases quantization difficulty, and PoT scaling then further amplifies the resulting error.
>
> Sec. 3.2/Fig. 2 show that rotation is highly effective in conventional INT4, but degrades in block quantization including BINT4/BFP4/MXINT4/MXFP4, revealing the incompatibility between global rotation and block quantization. Sec. 4.2/Fig. 4–5 show that global rotation spreads extreme outlier energy into many previously small-value channels, increasing the fraction of large values (activations above 1.5) from about 5% to 11%.  Based on this, the paper explicitly states that “global rotation amplifies the scales of regular blocks, thereby increasing their quantization difficulty.” Fig. 6 verifies, because regular blocks dominate in number, their accumulated error drives the overall degradation.
>
> The role of PoT is a further amplification, not the sole cause. Sec. 4.1/Fig. 3 show that MXFP4 quantization error increases with magnitude because PoT scaling is coarse at large scales; in Sec. 4.3, Proof 1 explains why regular-block scales increase after rotation, and Proof 2 plus the discussion below it explain why larger scales lead to larger quantization error, with additional PoT scale-rounding error further worsening the effect.
> > W2/Q2/Limitation: Trade-off.
>
> We agree block rotation has a clear limitation: by restricting mixing within each block, it is less effective for extreme outliers that would ideally be redistributed across more channels. The trade-off is between stronger global redistribution and better compatibility with fine-grained group-wise quantization. As shown in Fig. 4/Fig. 7 and discussed in Sec. 4.4, block-wise rotation can still reduce outlier concentration within a block, but can’t spread extreme outliers as broadly as global rotation. However, under MXFP4 this restriction is beneficial overall, because global rotation also spreads outlier energy into many regular blocks, inflating their scales and increasing the dominant error (Fig. 6). Tab. 20 and the new plots [**https://anonymous.4open.science/r/Sllq/**] make trade-off more explicit: at large quantization groups, global rotation still helps, but as the group becomes smaller, its benefit disappears and then reverses. Tab. 16 further shows that within block-wise rotation, too small a rotation block underuses redistribution capacity, while too large a block reintroduces cross-block interference; empirically, the best trade-off is achieved when rotation and quantization block sizes match.
> > Q3: Block-size matching.
>
> In our experiments, matching gives the most stable and best performance. Sec. 5.2 and App. Tab. 16 consistently support this. To relieve hyper-parameter tuning, we therefore view matching as the default choice in our setting, while agreeing that it is not a universal theorem for all formats or distributions.
> > W3: Although simple and practical, the solution is incremental.
>
> We agree BRQ is simple in form. However, the contribution is not merely the block-wise modification itself, but the systematic identification and explanation of a previously underexplored failure mode of rotation-based PTQ under MXFP4: Sec. 3 shows the instability of existing rotation methods in a unified MXFP4 setting. Sec. 4 shows that global rotation enlarges many regular-block scales, while MXFP4’s PoT scaling further amplifies the resulting error. BRQ is thus a direct correction derived from this analysis, rather than an arbitrary tweak.
>
> We believe the solution is practical and significant: MXFP4 is an increasingly relevant deployment format, while most existing W4A4/PTQ methods were developed for INT4-style settings and do not directly provide guidance for MXFP4. More broadly, we hope this analysis can guide future PTQ designs for MXFP4 and related grouped formats.
> > W4: Overhead.
>
> We clarify this at both the PTQ and inference stages. 1) PTQ: BRQ adds essentially no extra cost over QuaRot beyond switching to block-wise random Hadamard initialization; BRQ_{Spin} likewise adds no extra optimization or fusion overhead over SpinQuant. 2) Inference: block-wise rotation reduces online complexity from O(𝑁^2) to O(N×32). We added end-to-end results on RTX 5090 + LLaMA-3.2 1B: compared with QuaRot, BRQ reduces prefill latency by 2~6% and the extra overhead over native MXFP4 from 3.7%/7.4% to 1.5%/0.8%. Thus, BRQ does not increase PTQ cost and remains close to native MXFP4 efficiency.
> |Prefill(ms)| 256|512|
> |-|-|-|
> |BF16|100.60|201.77|
> |MXFP4|53.25|89.24|
> |QuaRot|55.20|95.85|
> |BRQ|54.03|89.91|

---

> > ### Author Rebuttal · Reviewer_Ke7E · 2026-04-04
> >
> > Thanks for the efforts in providing clarifications. I will maintain my overall score. While there exists concurrent work exploring block-based rotations and quantization, I find that this paper presents a more direct and practically grounded message, supported by clear empirical evidence, which sufficiently differentiates it in line with the authors’ claimed contributions.

---

> > > ### Author Response · Authors · 2026-04-04
> > >
> > > Thank you very much for the follow-up and for the thoughtful consideration of our rebuttal. We sincerely appreciate your recognition of the paper’s practical message and empirical support. Your feedback is very encouraging to us, and we are grateful for your time and constructive comments throughout the review process.

---

### Decision · Program_Chairs · 2026-04-30

**Decision:**

Accept (regular)

**Comment:**

With reviewer scores of 4, 4, 5, 5, and the overall assessment is positive, and I recommend acceptance. Reviewers agreed that the paper addresses a timely and practically important problem for MXFP4 quantization, and that its central insight is both clear and useful: global rotation interacts poorly with block-wise PoT scaling, whereas block-aligned rotation provides a simple and effective remedy. They also found the empirical study reasonably broad, with convincing gains over prior rotation-based baselines across multiple models and settings. In this sense, the paper makes a technically sound and practically relevant contribution that should be useful to researchers and practitioners working on low-bit LLM deployment.

The main concern raised across reviews is originality. Several reviewers viewed BRQ as a natural extension of existing rotation-based methods once the failure mode is identified, and noted that the originality lies more in the analysis and empirical clarification than in a fundamentally new algorithmic paradigm. Additional concerns were that the theoretical treatment is more intuitive than fully rigorous for MXFP4 specifically, and that the runtime evidence is somewhat limited and partly based on simulation rather than native FP4 hardware. That said, these concerns do not outweigh the paper’s strengths in soundness, clarity, and practical significance. I have read the rebuttal and reviewer discussion carefully and incorporated them into my decision; overall, I believe the paper makes a meaningful contribution and argues for acceptance.